# Patterns and associated factors of diabetes self-management: Results of a latent class analysis in a German population-based study

**Marcus Heise**[1]*, **Astrid Fink**[2], **Jens Baumert**[3], **Christin Heidemann**[3], **Yong Du**[3], **Thomas Frese**[1], **Solveig Carmienke**[1]

1 Institute of General Practice and Family Medicine, Medical Faculty of Martin Luther-University Halle-Wittenberg, Halle (Saale), Germany, 2 Institute of Medical Sociology, Medical Faculty of Martin Luther-University Halle-Wittenberg, Halle (Saale), Germany, 3 Department of Epidemiology and Health Monitoring, Robert Koch Institute, Berlin, Germany

* marcus.heise@medizin.uni-halle.de

**Data Availability Statement:** Legal restrictions concerning the participant's privacy prohibit us from publicly sharing our data set. Our manuscript

## Abstract

### Objective

Few studies on diabetes self-management considered the patterns and relationships of different self-management behaviours (SMB). The aims of the present study are 1) to identify patterns of SMB among persons with diabetes, 2) to identify sociodemographic and disease-related predictors of SMB among persons with diabetes.

### Research design and methods

The present analysis includes data of 1,466 persons (age 18 to 99 years; 44.0% female; 56.0% male) with diabetes (type I and II) from the population-based study German Health Update 2014/2015 (GEDA 2014/2015-EHIS). We used latent class analysis in order to distinguish different patterns of self-management behaviours among persons with diabetes. The assessment of SMB was based on seven self-reported activities by respondents (dietary plan, diabetes-diary, diabetes health pass, self-assessment of blood glucose, self-examination of feet, retinopathy-screenings and assessment of HbA1c). Subsequent multinomial latent variable regressions identified factors that were associated with self-management behaviour.

### Results

Latent class analysis suggested a distinction between three patterns of SMB. Based on modal posterior probabilities 42.8% of respondents showed an adherent pattern of diabetes self-management with above-average frequency in all seven indicators of SMB. 32.1% showed a nonadherent pattern with a below-average commitment in all seven forms of SMB. Another 25.1% were assigned to an ambivalent type, which showed to be adherent with regard to retinopathy screenings, foot examinations, and the assessment of HbA1c, yet nonadherent with regard to all other forms of SMB. In multivariable regression analyses, participation in Diabetes Self-Management Education programs (DSME) was the most

analyzed data of the nationwide population- based German Health Update (GEDA) 2014/2015 European Health Interview Survey (EHIS), conducted on behalf of the German Federal Ministry of Health by the Robert Koch Institute. The study protocol was inspected and approved by the German Federal Commissioner for Data Protection and Freedom of Information (reference number: III-401/008#0015). Written informed consent was obtained from all participants. Participants were informed about the goals and contents of the study, about privacy and data protection proceedings and their voluntary participation. These data cannot be shared publicly because informed consent from study participants did not cover public deposition of data. However, the minimal data set underlying the findings presented in this manuscript as well as the corresponding program files containing statistical analyses are archived in the Health Monitoring Research Data Centre at the Robert Koch Institute (RKI) and can be accessed by all interested researchers who meet the criteria for access to confidential data. Onsite access to the data set is possible at the Secure Data Center of the RKI's Health Monitoring Research Data Centre. Requests should be submitted to the Robert Koch Institute, Health Monitoring Research Data Centre, General-Pape-Straße 64, 12101 Berlin, Germany (e-mail: fdz@rki. de).

**Funding:** This work was supported by the Robert Koch Institute and the German Federal Ministry of Health, no specific grant number.

**Competing interests:** The authors have declared that no competing interests exist.

important predictor of good self-management behaviour (marginal effect = 51.7 percentage points), followed by attentiveness towards one's personal health (31.0 percentage points). Respondents with a duration of illness of less than 10 years (19.5 percentage points), employed respondents (7.5 percentage points), as well as respondents with a high socio-economic status (24.7 percentage points) were more likely to show suboptimal forms of diabetes self-management.

## Discussion

In the present nationwide population-based study, a large proportion of persons with diabetes showed suboptimal self-management behaviour. Participation in a DSME program was the strongest predictor of good self-management. Results underline the need for continual and consistent health education for patients with diabetes.

## Introduction

Diabetes mellitus is a significant global health issue affecting approximately 463 million people worldwide and 59 million people in Europe aged 20–79 years [1]. For persons with diabetes, consistent engagement in diabetes self-management can help achieving a near-normal blood glucose in order to reduce the risk of micro- and macrovascular complications and excess-mortality associated with diabetes [2, 3]. Therefore, national and international guidelines for the care of diabetes include self-management behaviour (SMB) as a core component in the treatment of diabetes [4]. Most components of SMB are necessary for both type I diabetes and type II diabetes, such as HbA1c-measurement, self-examination of feet, attendance of retinopathy-screenings and holding a diabetes pass. Furthermore, persons receiving insulin (both type I diabetes and type II diabetes) have similar recommendations regarding their SMB (such as the self-measurement of their blood glucose [SMBG] and keeping a diabetes-diary).

Due to the complexity of the concept of self-management, empirical studies use heterogeneous approaches for its operationalization. The "Summary of Diabetes Self-Care Activities" (SDSCA) measure [5], which represents one of the most frequently used inventories, constitutes an additive score of SMB (diet, exercise, SMBG, foot care, and medication within the preceding 7 days). The index is composed of the mean number of days per week on which the respondent behaved adherently, measured on a scale of 0 to 7. The "Self-Management-behaviour Index" (SMB-Index) proposed by Arnold-Wörner et al. [6] computes the sum of six dichotomized behaviours (physical exercise, foot care, SMBG, monitoring of body weight, diabetes diary, diet plan), forming a "compliance score". An analogue index is available in the form of the "revised Self-Care Inventory" (SCI-R) [7]. This instrument contains 15 items that inquire about the actual SMB behaviour in the last one to two months. This index is based (among other) on questions about medication, exercise, SMBG or the documentation of meals. To determine an overall self-treatment adherence score, the revised Self-Care Inventory calculates a mean based on these items ranging from 1 to 5. While all the aforementioned three indices inquire about factual self-management behaviour, the "Skills, Confidence and Preparedness Index" (SCPI) [8] mainly aims to provide a closer look at the subjective confidence in applying these forms of SMB. In this respect, the SCPI differs from other inventories. However, all four measurement instruments mentioned have in common that they use a tau-

equivalent measurement model when operationalizing SMB adherence, where all underlying items are added up in the form of an unweighted total score.

Based on these inventories, previous studies point to a heterogeneity in patients' self-management with the majority of patients showing a suboptimal adherence [9–11]. However, relatively few studies investigated the relationships and patterns of different aspects regarding self-management behaviours (SMB) of diabetes (e.g. Ruggiero et al. [12]; for patients with type I diabetes, see Mc Carthy et al. [13]; for paediatric type I diabetes see Rohan et al. [14]). From a behaviormetric point of view, composite indices may be questionable since various forms of SMB are not weighted and thus implied to be of equal relevance. Further, these additive indices neglect the underlying configural relations and co-occurrences between different components of SMB.

Furthermore, evidence on predictors of self-management behaviour is quite limited. Among other individual factors, a high level of education [15], living in a relationship [16] and female sex [17] are positively associated with patients' engagement in diabetes self-management. In contrast, financial constraints and a low socioeconomic status can impede patients' adherence to prescribed behavior [18]. Most importantly, diabetes self-management education (DSME) is relevant in the implementation of diabetes self-management [10] and was shown to be associated with a healthier lifestyle in routine healthcare [19].

However, to the best of our knowledge there is no empirical classification of different types of diabetes self-management addressing the co-occurrences of associated behaviours. Few studies have investigated patterns of SMB. Unarguably, there is a need to understand the connexion of various forms of SMB and to identify associated factors. Furthermore, evidence on predictors of SMB is limited, especially in nationwide population-based studies. Ultimately, additional information on possible predictors of SMB might help to tailor targeted and appropriate interventions, thus improving self-management in patients with diabetes. Therefore, the aims of the present study are:

1. to identify patterns of SMB among persons with diabetes,

2. to identify sociodemographic and disease-related predictors of SMB among persons with diabetes.

## Materials and methods

### Study design and population

The German Health Update (GEDA) 2014/2015 integrated modules of the European Health Interview Survey (EHIS) wave 2 and was conducted by the Robert Koch Institute on behalf of the German Federal Ministry of Health between November 2014 and July 2015. The survey used self-administered questionnaires provided either as online or paper-based versions and yielded a response rate of 26.9% [20]. Details of participant sampling and questionnaire design are described elsewhere [20, 21].

The German Federal Commissioner for Data Protection and Freedom of Information approved the corresponding study (reference number: III-401/008#0015). All respondents gave their written informed consent. Participants were informed about the goals and contents of the study, about privacy and data protection proceedings, and that their participation in the study was voluntary.

### Inclusion/Exclusion criteria

Inclusion criteria for GEDA 2014/2015-EHIS were a permanent residence in Germany and age ≥ 15 years. Our analysis is limited to survey participants reporting diabetes in the past 12

months and aged $\geq$ 18 years (n = 1,712). Five women with current or recent gestational diabetes were excluded. Further, a number of 241 respondents with missing sociodemographic or disease-related information were removed from the analysis by list-wise deletion. The resulting net sample consisted of 1,466 respondents. With regard to the single items of SMB, we could not identify any significant differences between excluded respondents and the net sample of complete cases (see S1 Table). With this background, arguments can be made against the imputation of missing values in exogenous variables [22]. For this reason, the following analyses are limited to complete cases.

## Assessment of SMB

The outcome of our analysis are patterns of SMB. Within latent class analysis, we defined this comprehensive construct as a categorical variable. As its indicators, we used participants' self-reported frequency in regard to seven forms of SMB: Currently keeping to a dietary plan for their diabetes (Yes / No) or a diabetes-diary (Yes / No), ever keeping a diabetes health pass (Yes / No), SMBG performed by oneself or by relatives (No, daily, number per week or month; categorized as "at least once a month" / "less frequently"), self-examination of feet (no, daily, number per week or month; categorized as "daily or occasional" / "never"), retinopathy screenings (times within last 12 months; categorized as "at least once within last 12 months" / never within last 12 months) and assessment of HbA1c (times within last 12 months; categorized as "at least 4 times within last 12 months" / "less than 4 times within last 12 months"). Even though partial metric scale responses were available for some items, this information was missing for some respondents due to erratic response behaviour. The skewness of the distributions, avoidance of sparse data and otherwise limited comparability between the seven SMB indicators made it necessary to dichotomise these variables. The exact phrasings of the mentioned items within the GEDA-questionnaire are presented in S2 Table. The questions included in the GEDA survey relate to factual self-management behaviour and are in this respect similar to the aforementioned operationalisations within the SDSCA, SMB-Index or SCI-R. However, some of the items in the GEDA study use heterogeneous time references ("currently" in the case of dietary plan or diabetes-diary; "last 12 months" in the case of retinopathy-screenings or assessment of HbA1c).

## Sociodemographic and disease-related variables

The selection of covariates was based on the availability of data in the GEDA survey and on central findings of previous studies. Among sociodemographic factors of SMB, age [12, 23], sex [17], socioeconomic status [15, 18], employment [15], and partnership status [16] have been reported in the literature. Among disease-related variables, time since diagnosis [15], DSME-participation [10, 19], limitation due illness [24] and attendance toward health [15, 21] emerged as factors of SMB. We adopted these variables as exogenous covariates within our analyses. However, the selection of these covariates was not based on a systematic literature review. Thus, our analysis does not aim at a systematic comparison of theories, but rather has an explorative character, which is primarily due to the availability of the data within the present survey.

Sociodemographic variables comprised sex (male/female), age (in years), socioeconomic status (SES) based on a quasi-metric SES-score (3 to 21 points) [25], currently living together with a partner or as couple (yes / no) and occupational status (employed / unemployed or retired or unable to work). It should be noted that in case of unemployment or retirement, the SES index does not include the *current* employment status, but uses the respondents' last professional activity instead for its calculation. Furthermore, the SES index also refers to the level

of education and income and thus differs from the occupational status. No evidence of multi-collinearity between the two variables could be identified. For this reason, we use the SES index and the occupational status as two separate factors.

Disease-related characteristics included limitations due to illness that lasted at least 6 months (high or moderate / none), patients' attentiveness towards their own health (Likert score ranging from 1 points for "no attendance" to 5 points for "very high"), ever-participation in a DSME program (yes / no) and time since diagnosis of diabetes ($< 10$ years / $\geq 10$ years).

## Statistical analysis

Descriptive statistics with unweighted absolute frequencies, weighted relative frequencies, means and standard deviations were used to describe the study population. Calculation of corresponding weighting factors is based on sampling points. These weighting factors refer to the complex multistage sampling design of the GEDA 2014/2015-EHIS survey and address a potential sample-selection bias [20, 21]. In order to identify similarities between respondents regarding their diabetes SMB and to classify different degrees of adherence, we used latent class analysis (LCA) [26]. The procedure of LCA constitutes a particular type of a finite mixture model. It represents a person-centred approach to statistical modelling of unobserved population heterogeneity based on manifest indicators. In the present analysis, we used LCA as a data-driven approach to identify mutually exclusive and exhaustive patterns of SMB. The seven aforementioned items regarding self-management behaviours served as indicators of a categorical latent variable.

The decision on the number of latent classes within our study was primarily based on the principle of parsimony and the interpretability of its partitioning [27]. In addition, model selection took into account penalized likelihood criteria, namely Akaike-Information-Criterion (AIC) [28], Bayes-Information-Criterion (BIC) [29] and sample-adjusted Bayes-Information-Criterion (ABIC) [30], without rigidly following these criteria at the expense of interpretability [26, 27].

Content-related interpretations of the latent classes were based on conditional prevalences of the indicator-variables [26]. Frequency distributions of the latent classes were estimated based on the modal posterior probabilities (most likely latent class membership). We used multinomial latent variable regression [31] in order to predict the posterior probabilities of the latent classes (i.e. the conditional probability based on LCA-model) by the nine aforementioned sociodemographic and disease-related variables. To prevent a potential misspecification of our model, analyses began with nine sequentially, bivariate regressions, limited to one covariate at a time. Covariates with statistically significant ($\alpha = 0.05$, see below) or substantial effects (discrete change in absolute probabilities $\geq 10$ percentage points) within those bivariate regressions were thereafter included within a final, multivariate model. However, a changing combination of exogenous covariates within finite mixture models may also alter the composition of latent classes themselves, thus hindering content-related interpretations of the latent classes [32]. To address this concern, we fixed the starting parameters of the latent class measurement model and used a Manual ML-three-step approach [33] for correlational analyses of the sociodemographic and disease-related factors. Evaluation of the strength and relevance of various predictors was based on marginal effects [34] (i.e. changes in the predicted posterior probabilities). We plotted predicted posterior probabilities for covariates to further illustrate relevant effects. All LCA models used weighting factors (see above). An additive index was also calculated from the dichotomized SMB indicators. In weighted linear regressions, this index value was predicted by the sociodemographic and disease-related variables. The corresponding findings on relevant factors were compared with the results of multinomial latent variable regressions.

An alpha level of 0.05 was used as a threshold for statistical significance. All exogenous covariates were inspected for collinearity (or sparse data in the case of categorical variables) before analysis.

All SMB investigated in this manuscript are covered by the current diabetes therapy and DSME guidelines [35, 36] with two minor exceptions: One, SMBG is recommended only for those persons who receive insulin treatment or oral antidiabetic medication with risk of hypoglycaemia or are in a potential instable metabolic situations (e.g. newly diagnosed diabetes, metabolic aggravation, surgery, infection, aggravation of diabetic metabolic situation [35, 36]). German health insurance does not pay SMBG for patients with diabetes not fulfilling the above named criteria [37]. It can be assumed that some patients not fulfilling reimbursement criteria choose to perform SMBG on their own costs irregularly. Within another sample [38], 74,8% of persons with type 2 diabetes received no insulin as a treatment and therefore did not perform SMBG regularly. Therefore, we repeated the LCA within a sensitivity analysis excluding SMBG as an indicator variable. Second, therapy and DSME guideline [35] recommend to develop a dietary concept for DSME participants, but not strictly a dietary plan. The use of strict diet plans in routine clinic care for therapy of diabetes has decreased over the last years. Therefore, a second sensitivity analysis excluded dietary plan as an indicator variable of the latent classes.

All analyses were performed using Stata v.16.1 [39] and MPLUS Version 6.11 [40].

## Results

### Descriptive statistics

Sample characteristics are presented in Table 1. Our sample consisted of 56.0% men and 44.0% women with an average age of 65.5 years (range = 19 to 99 years). About two thirds of respondents consisted of pensioners (63.9%) and lived together with their partner or as a couple (70.7%). More than half of our respondents have already participated at least once in a DSME program (62.7%) at least once. The median for the time since the (first) diagnosis of diabetes was 9 years ($<$ 10 years: 52.9%; $\geq$ 10 years: 47.1%). The most frequent self-reported SMB were retinopathy-screenings (75.5%), self-examination of feet (71.7%) SMBG (66.5%). The least frequently reported SMB were currently following a dietary plan (13.3%) and keeping a diabetes (management) diary (36.4%). Only 40.5% of our sample had four assessments of HbA1c within the past twelve months.

### Latent-classes as patterns of SMB

The results of a series of unconstrained, weighted latent-class models are presented in Table 2. While AIC and ABIC suggest a partitioning of four to five latent classes, BIC points towards a model with three latent classes. Fig 1 illustrates the corresponding conditional prevalences of SMB for these three classes.

In analyses presented as supplemental figures (see S1 and S2 Figs), a partitioning into four or five classes did not provide any added value in terms of interpretability. In the 4-class solution, the adherent and non-adherent classes are confirmed. The mixed-pattern, on the other hand, differentiates into two subtypes: One with an above-average engagement in SMBG and diabetes diary; and a second subtype with a comparatively higher frequency in HbA1c assessments. However, regarding the other indicator variables, the two mixed-subtypes in S1 Fig are pretty similar. In order not to over interpret these weak differences and because one of the subtypes has a low a-posteriori probability (n = 148), we suggest that a four-class solution (with two subtypes of an ambivalent SMB-pattern) offers no added value compared to the three-class solution. The low frequency of class 1 (n = 23) within the 5-class solution most likely

**Table 1. Characteristics of the study population of 1,458 persons with diabetes in the past 12 months (GEDA 2014/2015-EHIS).**

| | n/n$_{valid}$[a] | %[b] | M (SD)[c] |
|---|---|---|---|
| Sociodemographic and disease-related factors | | | |
| Sex | | | |
| *Male* | 851 / 1,466 | 56.0% | |
| *Female* | 615 / 1,466 | 44.0% | |
| age (years) | 1,466 | | 65.5 (13.6) |
| SES-score [3 to 21] | 1,466 | | 10.7 (3.7) |
| living together with spouse or as couple | | | |
| *No* | 411 / 1,466 | 29.3% | |
| *Yes* | 1055 / 1,466 | 70.7% | |
| occupational status | | | |
| *employed person* | 431 / 1,466 | 28.7% | |
| *unemployed person* [d] | 98 / 1,466 | 7.4% | |
| *retired / unable* | 937 / 1,466 | 63.9% | |
| limitation due to illness within last 6 months | | | |
| *High* | 268 / 1,466 | 18.7% | |
| *moderate* | 483 / 1,466 | 33.1% | |
| *None* | 715 / 1,466 | 48.2% | |
| attendance toward health [1 to 5] | 1,466 | | 3.6 (0.8) |
| participation in DSME program | | | |
| *No* | 547 / 1,466 | 37.3% | |
| *Yes* | 919 / 1,466 | 62.7% | |
| time since diagnosis of diabetes | | | |
| *< 10 years* | 782 / 1,466 | 52.9% | |
| *≥ 10 years* | 684 / 1,466 | 47.1% | |
| Self-management behaviours | | | |
| currently keeping dietary plan | | | |
| *No* | 1,268 / 1,462 | 86.7% | |
| *Yes* | 194 / 1,462 | 13.3% | |
| currently keeping diabetes-diary | | | |
| *No* | 952 / 1,463 | 63.6% | |
| *Yes* | 511 / 1,463 | 36.4% | |
| ever kept diabetes health pass | | | |
| *No* | 781 / 1,461 | 52.6% | |
| *Yes* | 680 / 1,461 | 47.4% | |
| self-measurement of blood glucose (SMBG) | | | |
| *Less than once a month* | 499 / 1,463 | 33.5% | |
| *At least once a month* | 964 / 1,455 | 66.5% | |
| self-examination of feet | | | |
| *never* | 408 / 1,436 | 28.3% | |
| *daily or occasionally* | 1,028 / 1,436 | 71.7% | |
| retinopathy-screenings within last 12 months | | | |
| *never within last 12 months* | 349 / 1,457 | 24.5% | |
| *at least once within last 12 months* | 1,108 / 1,457 | 75.5% | |
| assessment of HbA1c | | | |
| *less than 4 times within last 12 months* | 598 / 1,432 | 40.5% | |
| *at least 4 times within last 12 months* | 834 / 1,432 | 59.5% | |

This table shows absolute frequencies and estimated prevalences in percent based on weighting factors.

[a] = We show unweighted absolute frequencies (n/nvalid)

[b] = Percentages (%) are calculated with reference to overall cohort including weighting factors

[c] = means and standard deviations take weighting factors into account

[d] = the category of "unemployed person" includes pupils, students and homemakers.

Abbreviations: DSME–structured education program for patients with diabetes mellitus, HbA1c –haemoglobin A1c; M ± SD–mean ± standard deviation, n$_{valid}$–cases with non-missing values; SES- socioeconomic status.

**Table 2. Information criteria (penalized likelihood criteria) for a series of weighted latent-class models without covariates (unconditional models).**

| number of latent classes | Log-Likelihood | AIC | BIC | ABIC |
|---|---|---|---|---|
| Independence | -6108.3 | 12230.7 | 12267.7 | 12245.5 |
| 2 | -5690.9 | 11411.8 | 11491.2 | 11443.5 |
| 3 | -5650.5 | 11346.9 | **11468.6** | 11395.5 |
| 4 | -5621.5 | 11305.0 | 11469.0 | **11370.5** |
| 5 | -5609.3 | **11296.5** | 11502.9 | 11379.0 |

n = 1466; all models took weighting factor in account

Abbreviations: AIC–Akaike-Information-Criterion; BIC–Bayes-Information-Criterion; ABIC: Sample-adjusted Bayes-Information-Criterion.

indicates sample-specific idiosyncrasies. Likewise, the high extent of boundary estimates (i.e. health pass, diabetes diary, dietary plan) suggests that the 5-class solution represents a model-misspecification. To achieve a parsimonious model with as few latent classes as possible and clear interpretability, we chose a distinction between three types of SMB-patterns. These three emergent latent classes in Fig 1 can be characterized as follows:

**Adherent pattern of diabetes self-management.** An "adherent self-management pattern" is characterized by frequent self-assessment of SMBG (93.8%), regular examinations of the feet

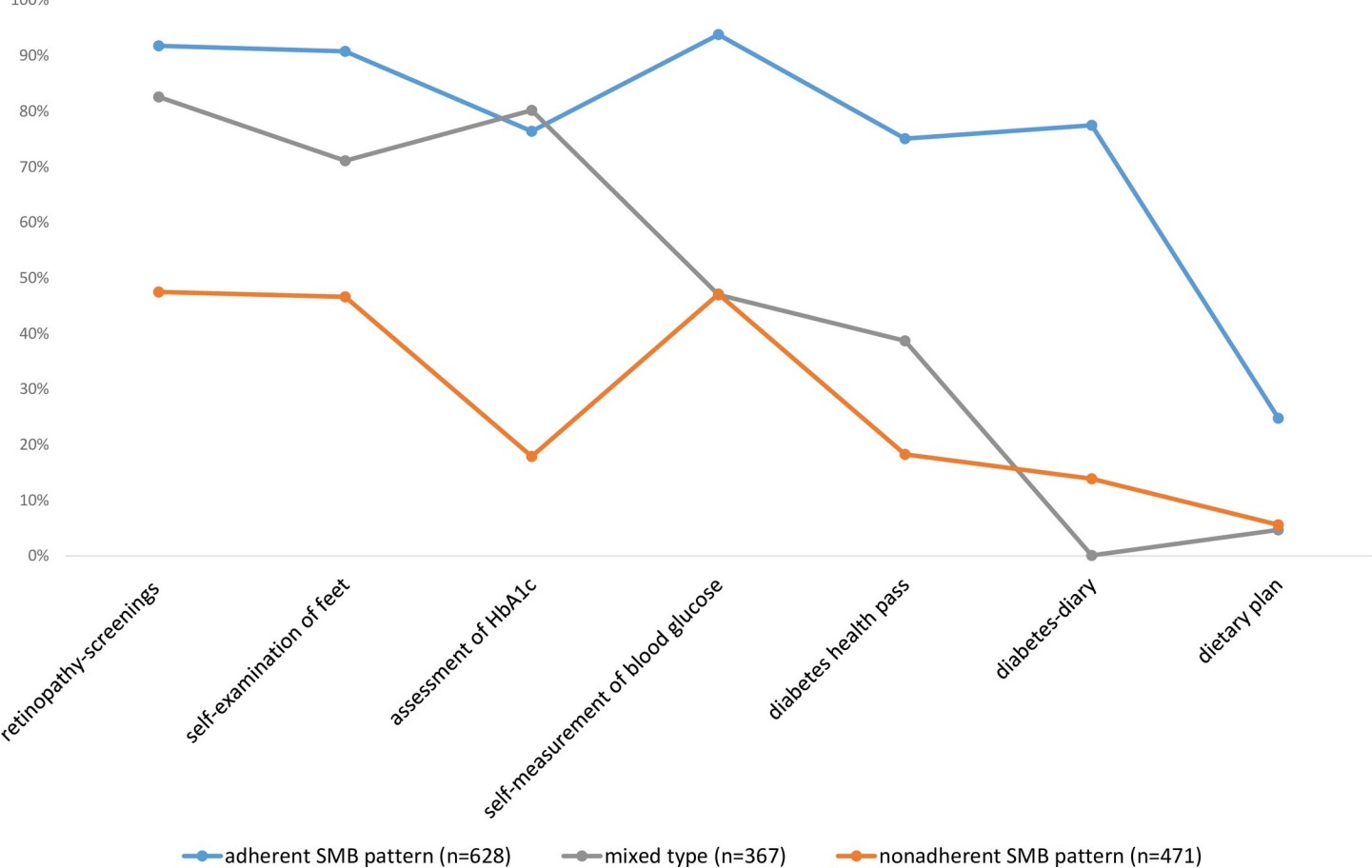

**Fig 1. Profiles of the latent self-management classes in the 3-group model with 7 indicator variables.** Conditional prevalences, n = 1,466.

(90.8%), retinopathy-screenings (91.8%), quarterly assessment of HbA1c (76.4%) as well as keeping a diabetes health pass (75.1%) and a diabetes-diary (77.5%). Respondents within this class are more likely than those within the other two classes to follow a dietary plan (24.8%). Based on average posterior probabilities, 42.8% (n = 628) of our respondents belong to the class of an "adherent" diabetes self-management pattern.

**Nonadherent pattern of diabetes self-management.**   In contrast, the latent class of "*non-adherent* self-management pattern" shows low frequency in these seven forms of SMB (retinopathy-screenings: 47.5%; SMBG: 47.1%; feet examination: 46.6%; assessment of HbA1c: 17.9%; diabetes health pass: 18.3%; diabetes-diary: 13.9%; dietary plan: 5.6%). Based on average posterior probabilities, 32.1% (n = 471) of our respondents show a nonadherent form of diabetes self-management.

**Mixed type.**   Furthermore, a mixed type emerges, which can be located between the two classes of "adherent" and "nonadherent" diabetes self-management. This mixed type shows to be adherent regarding retinopathy screenings (82.6%), foot examinations (71.1%) and the assessment of HbA1c (80.2%). Regarding the other forms of SMB, however, the mixed-type is similar to the pattern of "nonadherent" diabetes self-management (SMBG: 47.0%; diabetes health pass: 38.7%; diabetes-diary: 0.1%; dietary plan: 4.7%). Based on modal posterior probabilities, 25.1% (n = 367) of our respondents can be classified into a "mixed type" with ambivalent frequency of SMB.

Two sensitivity analyses excluded the indicator variables "SMBG" or "keeping a dietary plan", respectively. The penalized likelihood criteria do not contradict the previous distinction between three SMB patterns (see S3 and S4 Tables). Furthermore, the three latent classes within both sensitivity analyses lead to similar interpretations compared to the model which was based on all seven indicators of SMB (see S3 and S4 Figs).

## Prediction of SMB patterns by sociodemographic and disease-related factors

Table 3 contains results of a series of multinomial latent variable regressions, in which the latent classes were predicted by sociodemographic and disease-related factors. The first nine models constitute bivariate regressions containing one exogenous factor at a time. The corresponding marginal effects of these bivariate regressions are illustrated in Fig 2.

Within bivariate analyses, the participation in a DSME program emerged as the strongest predictor of SMB patterns. As illustrated in Fig 2, participants of a DSME program showed an average posterior probability for adherent self-management of 61.0%, whereas this probability was only 8.5% for non-participants. This led to a marginal effect (i.e. change in posterior probabilities) of 52.5 percentage points. Opposing this information, the average posterior probability for a nonadherent SMB pattern was substantially higher among non-participants compared to DSME-participants (58.8% vs. 14.6%; with a marginal effect of 44.2%). These effects of DSME participation were also significant in a statistical sense (p<0.001). The attentiveness towards one's personal health constituted another substantial and significant predictor of SMB and was positively associated with an adherent pattern. As illustrated in Fig 2, the average posterior probability for an adherent SMB pattern was 24.5% for respondents who pay no attention towards their own health (z-score = -3.1, untransformed value of "1"). For respondents, who are very attentive towards their own health (z-Score = 1.8, untransformed value of "5"), this probability rose to 50.7%, netting a marginal effect of 26.2 percentage points. Conversely, high attendance toward own health reduced the average posterior probability for a nonadherent SMB pattern from 51.7% to 21.5%, netting a marginal effect of 30.2% percentage points. These bivariate effects of attentiveness toward one's own health reached statistical significance

**Table 3. Results of multinomial latent variable regressions: Posterior probabilities predicted by sociodemographic and disease-related factors (logistic slopes and marginal effects).**

| Modell | | logistic slope: mixed-type vs non-adherent SMB pattern | | logistic slope: adherent SMB pattern vs nonadherent SMB pattern | | marginal effects (average change or discrete change) [a] | | |
|---|---|---|---|---|---|---|---|---|
| | | β | p | β | p | non adherent SMB pattern | mixed-type | adherent SMB pattern |
| Modell 1 | ever-participation in DSME program (vs. never) | 1.100 | <0.001 | 3.367 | <0.001 | -44.2% | -8.3% | 52.5% |
| | Intercept | -0.584 | | -1.937 | | | | |
| Modell 2 | age (z-score) | 0.247 | 0.091 | -0.040 | 0.685 | -8.2% | 29.1% | -20.9% |
| | Intercept | -0.125 | | 0.281 | | | | |
| Modell 3 | female (vs. male) | 0.428 | 0.124 | 0.319 | 0.106 | -7.7% | 4.9% | 2.8% |
| | Intercept | -0.305 | | 0.153 | | | | |
| Modell 4 | SES-Score (z-Score) | -0.249 | 0.075 | -0.303 | 0.002 | 27.2% | -7.2% | -20.0% |
| | Intercept | -0.132 | | 0.263 | | | | |
| Modell 5 | living together (vs. not living together) | -0.124 | 0.692 | -0.093 | 0.672 | 2.2% | -1.4% | -0.8% |
| | Intercept | -0.033 | | 0.352 | | | | |
| Modell 6 | high/moderate limitation due illness (vs. none) | 0.204 | 0.457 | 0.432 | 0.025 | -7.3% | -0.8% | 8.1% |
| | Intercept | -0.219 | | 0.064 | | | | |
| Modell 7 | attendance toward health (z-Score) | 0.211 | 0.147 | 0.328 | 0.002 | -30.2% | 4.0% | 26.2% |
| | Intercept | -0.099 | | 0.298 | | | | |
| Modell 8 | employed (vs. unemployed / retired / unable) | -0.669 | 0.029 | -0.368 | 0.068 | 10.7% | -8.8% | -1.9% |
| | Intercept | 0.071 | | 0.402 | | | | |
| Modell 9 | time since diagnosis ≥ 10 years (vs. < 10 years) | 1.015 | 0.001 | 1.469 | <0.001 | -25.8% | 2.9% | 22.9% |
| | Intercept | -0.502 | | -0.348 | | | | |
| multivariate Regression | ever-participation in DSME program (vs. never) | 1.148 | 0.000 | 3.406 | <0.001 | -43.9% | -7.9% | 51.7% |
| | SES-Score (z-Score) | -0.204 | 0.197 | -0.329 | 0.018 | 24.7% | -2.9% | -21.8% |
| | attendance toward health (z-Score) | 0.125 | 0.411 | 0.361 | 0.013 | -13.6% | -17.3% | 31.0% |
| | employed (vs. unemployed / retired / unable) | -0.456 | 0.216 | -0.251 | 0.421 | 7.5% | -7.2% | -0.3% |
| | time since diagnosis ≥ 10 years (vs. < 10 years) | 0.779 | 0.015 | 1.137 | <0.001 | -19.5% | 3.5% | 16.0% |
| | Intercept | -0.756 | | -2.396 | | | | |

n = 1466; notes: a) marginal effects for categorical, dichotomous predictors refer to average differences in predicted posterior probabilities between the two possible values of the covariate (discrete change); marginal effects for metric predictors refer to the average change in predicted posterior probabilities when covariate changes from minima to maxima; latent classes parameters fixed within Manual ML-three-step approach.

Abbreviations: DSME–structured education program for patients with diabetes mellitus, SES- socioeconomic status.

(p = 0.002). Time since diagnoses of diabetes was a further relevant and significant predictor (p<0.001). Respondents with a diabetes duration below median (10 years) had an above-average probability regarding a nonadherent SMB pattern (43.3% vs. 17.4%). In contrast, respondents with a duration of illness of ten years or more were more likely to show an adherent SMB pattern (53.5% vs. 30.6%). Occupational status constituted a comparatively weaker–yet still significant (p = 0.029)–predictor of the latent classes, with marginal effects of up to 10.7 percentage points. Among employed respondents, the nonadherent pattern (38.7% vs. 28.0%) was more strongly represented while non-employed persons with diabetes were comparatively more likely to show a mixed type of self-management (30.1% vs. 21.3%). Respondents with a high or moderate limitation due to illness within the last six months had an above-average probability for an adherent SMB pattern compared to other respondents (45.3% vs. 37.2%;

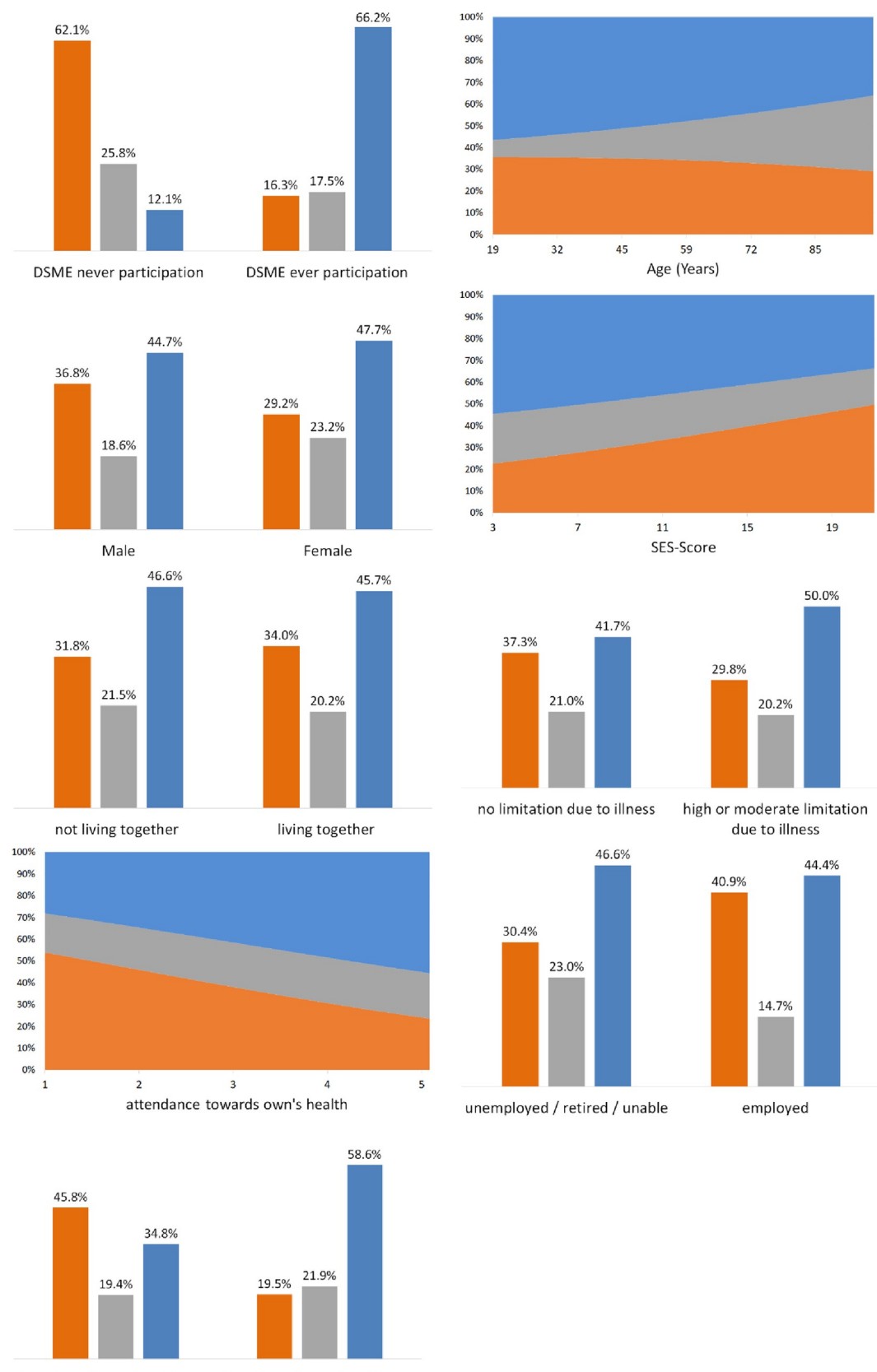

**Fig 2. Predicted posterior probabilities of the latent self-management classes by sociodemographic and disease-related factors within bivariate, multinomial latent variable regressions.** n = 1466, blue = adherent SMB pattern (n = 628); grey = mixed-type (n = 367); orange = nonadherent SMB pattern (n = 471) Abbreviations: DSME–structured education program for patients with diabetes mellitus, SES- socioeconomic status. All predicted probabilities according to the coefficients stated in Table 3, assuming other covariates are fixed at sample-specific means.

p = 0.025). A last significant effect within bivariate analyses concerns respondents with a low socioeconomic status, who were more likely to show an adherent pattern of SMB compared to respondents with a high socioeconomic status (49.6% for "low" vs. 29.7% for "high"). Accordingly, a high socioeconomic status was positively associated with a nonadherent form of SM among our participants (p = 0.002).

The other three predictors were merely weakly and non-significantly associated with the latent classes. For instance, no substantial sex effects are evident regarding SMB patterns. Age was negatively associated with both the nonadherent and adherent patterns and was positively associated with a mixed type. However, this effect did not achieve statistical significance. Respondents who were living together with their spouse or as couple do not differ from other respondents regarding their SMB patterns.

The multivariate model shown in Table 3 was limited to the factors that showed significant associations to the latent classes within the previous regressions. Note that the exogenous covariate "limitation due to illness" failed to achieve practical and statistical significance within multivariate analysis and was therefore excluded from the final model. Although occupational status also failed to achieve statistical significance, this covariate remained in the final model since the corresponding marginal affects pointed towards a clinically relevant association.

All effects in the final model remain significant and replicate the associations that already emerged in the previous, bivariate analyses. The multivariate model confirms participation in a DSME program as the strongest predictor of an adherent SMB pattern (p < 0.001). Attendance towards personal health (p = 0.013) and time since diagnosis (p < 0.001) are still positively related to an adherent SMB pattern, while socioeconomic status (p = 0.018) and employment (n.s.) are associated with a nonadherent SMB pattern.

Excluding the two indicator variables "SMBG" and "keeping a dietary plan" within two subsequent sensitivity analysis largely replicated the above findings (see S5 and S6 Tables).

## Comparison to results of linear regressions based on an additive index

The aim of the following analyses is a comparison between the previous results of the multinomial latent variable regressions and those results using an additive measure of SMB-adherence. For this purpose, the seven dichotomized SMB indicators were combined into an additive index with a value range between 0 (low adherence) to 7 (high adherence). Based on biserial correlations, this index achieved a suboptimal psychometric reliability of Cronbach's $\alpha$ = 0.62. Note that 77 additional cases had to be excluded due to missing values. The corresponding results of weighted linear regressions are presented in Table 4.

Bivariate Regressions mainly reproduce the effects of multinomial latent variable regressions from Table 3. DSME participation (b = 1.72) and time since diagnosis (b = 1.00) are confirmed as the most important, positive factors of adherence in SMB of diabetes. The effects of SES-Score (b = -0.21), attendance towards personal health (b = 0.28) and occupational status (b = -0.41) reach statistical significance, yet fail to reach a level of practical significance with respect to their corresponding proportions of explained variance ($R^2 \leq$ 2%). In contrast to the results of the LCA, sex (b = 0.26), age (b = 0.05) and limitation due illness (b = 0.45) emerge as significant predictors of the additive SMB-Index in bivariate analyses, but these factors also fail to reach a level of practical significance given their low $R^2$. One explanation for these

**Table 4. Results of weighted linear regressions: Additive SMB-Index predicted by sociodemographic and disease-related factors (unstandardized slopes).**

| | | unstandardized slope | | $R^2$ |
|---|---|---|---|---|
| **Modell** | | **b** | **p** | |
| Modell 1 | ever-participation in DSME program (vs. never) | 1.72 | <0.001 | 0.215 |
| | Intercept | 2.63 | | |
| Modell 2 | age (z-score) | 0.05 | 0.407 | < 0.01 |
| | Intercept | 3.71 | | |
| Modell 3 | female (vs. male) | 0.26 | 0.019 | < 0.01 |
| | Intercept | 3.59 | | |
| Modell 4 | SES-Score (z-Score) | -0.21 | <0.001 | 0.012 |
| | Intercept | 3.69 | | |
| Modell 5 | living together (vs. not living together) | -0.09 | 0.535 | 0.016 |
| | Intercept | 3.77 | | |
| Modell 6 | high/moderate limitation due illness (vs. none) | 0.45 | <0.001 | 0.024 |
| | Intercept | 3.47 | | |
| Modell 7 | attendance toward health (z-Score) | 0.28 | <0.001 | 0.02 |
| | Intercept | 3.71 | | |
| Modell 8 | employed (vs. unemployed / retired / unable) | -0.41 | 0.002 | 0.011 |
| | Intercept | 3.83 | | |
| Modell 9 | time since diagnosis ≥ 10 years (vs. < 10 years) | 1.00 | <0.001 | 0.078 |
| | Intercept | 3.23 | | |
| Modell 10 | ever-participation in DSME program (vs. never) | 1.59 | <0.001 | 0.279 |
| | SES-Score (z-Score) | -0.16 | 0.001 | |
| | attendance toward health (z-Score) | 0.21 | <0.001 | |
| | employed (vs. unemployed / retired / unable) | -0.24 | 0.040 | |
| | time since diagnosis ≥ 10 years (vs. < 10 years) | 0.58 | <0.001 | |
| | Intercept | 2.49 | | |

n = 1,389; the additive SMB-Index ranges from 0 points (minimum) to 7 points (maximum).

Abbreviations: DSME–structured education program for patients with diabetes mellitus, SES- socioeconomic status.

deviations is the fact that the additive index does not differentiate between a mixed-type and nonadherence, but instead merges these two patterns and conceptualizes adherence as a linear, one-dimensional metric. The results of the multivariate regressions are largely consistent with the preceding analyses, with one major exception. In the linear regressions, occupational status was found to have a significant negative effect on adherence, whereas in the LCA it contributed (non-significantly) to a shift between mixed and nonadherent patterns.

## Discussion

This paper presented an empiric grouping of persons with diabetes based on various aspects of their diabetes self-management. Without relying on an operationalization solely based on single item indicators or on unweighted composite indices, this empiric grouping identified three mutually exclusive patterns of diabetes self-management. Only 42.8% of our respondents showed adherence in their self-management of diabetes. Our results therefore concur with other studies stating that a large portion of patients with diabetes show suboptimal forms of SMB [9–11]. Based on this empiric grouping, our population-based study presents and confirms an array of demographic and disease-related predictors of self-management for persons with diabetes. Participation in a DSME program was the strongest predictor of good self-management.

Although the survey data we used are five years old, our findings are still relevant. DSME programs, their curricula, aims and content have not changed in the last five years. However, new medications emerging as SGLT2 inhibitors, have been integrated in the medication education sessions. For specific new technical devices, e.g. the application and use of continuous subcutaneous SMBG, a new DSME program for educational purposes been developed. However, this latter program is used as an addition to the other DSME. Also, in the current pandemic, Online-DSME formats have emerged to ensure some stability of care. Online-DSME formats have emerged to ensure some stability of care. However, they are not widely applied, as 74.7% of registered DSME trainers reported not ever having performed an Online-DSME and educate the same curriculum as the "vis-à-vis" DSME [41]. Furthermore, we are of the opinion that our findings are of high relevance, since to best of our knowledge the present study is the only nationwide survey investigating SMB and DSME attendance in Germany. By employing Latent Class Analysis, we grouped respondents based on their adherence in self-management behaviour. These latent classes correspond to the general willingness to engage and participate in the treatment of diabetes and are merely intermediately connected to specific medications or guidelines.

## Main findings compared to other population-based studies

International literature underlines the effectiveness of DSME for effective diabetes SMB [10, 19] and concurs that participation in DSME is positively associated with glycaemic control [3]. In accordance, participation in DSME emerges as the strongest factor of an adherent SMB pattern within our analysis. However, based on average posteriori probabilities, 12.1% of those not having participated in a DSME program are classified as being "adherent". Thus, non-participants can also be adherent in their SMB and instead might use alternative ways to inform themselves about coping with diabetes or might receive extensive support from the practice personnel. In addition, many reasons exist for (adherent) respondents to decide against DSME training. In particular, truck drivers or commuters may not be able to attend a DSME program due to work-related reasons. Nevertheless, our results concur with the literature that persons with diabetes receiving DSME are more likely to foster adherence to recommended self-management activities [10, 15]. Therefore, our results indicate that people with diabetes should be informed by their general practitioner about the possibility of DSME training. However, empirical evidence suggests that a one-time participation in a DSME program may not yield long-term effects [42]. This emphasizes the importance of repeated DSME participation for persons with diabetes [4]. Yet, only 62.7% of respondents within our sample participated in a DSME-program at least once in their life. In other assessments, comparable participation rates of 50% to 53% for Germany are reported [10]. Given the strong effects of DSME participation on frequent and adherent SMB, it would be imperative to increase the scale and scope of such education-programs.

Various studies [15, 21] found that the resolution to prevent or reduce the risk of developing diabetes complications improves the determination to engage in self-management. This corresponds to our effects regarding mindfulness towards patients' own health, which showed to be another relevant facilitator of a good diabetes self-management. In our multivariate analysis, the attendance towards one's personal health emerged as a separate effect which was independent from DSME participation.

Prior research showed that employment has a negative effect on SMB among patients with diabetes [15, 43]. Especially the prioritization of work over medical issues can be deterrent to maintaining a healthy lifestyle and adherent self-management [44]. Employed patients with diabetes may lack the time and energy to follow a dietary plan, to be physically active or to

perform SMBG on a regular basis [43]. Our results concur that occupation can be a barrier towards an efficient SMB. For this reason, DSME-programs should be tailored to meet the specific needs of employed patients. Furthermore, clinicians should continuously consult the patients in how to integrate SMB in their daily life and ensure a regular follow up of their employed patients.

Contrary to our results, numerous studies found evidence that a high level of education [43, 45, 46] and high socioeconomic status [15, 18, 47, 48] are facilitators of adherence in diabetes self-management. This inverse effect of socioeconomic status on self-management within the present data is not a statistical artefact produced by multivariate analysis, since zero-order bivariate correlations within single item analyses confirm this association. We cannot offer an exhaustive explanation why our results are in contradiction to the literature. The specifics of the German statutory health insurance system might constitute a plausible explanation. Our results also concur with qualitative studies among persons with diabetes, in which respondents with a low socioeconomic status tended to follow SMB instructions strictly and almost literally, whereas persons with high socioeconomic status interpreted SMB instructions rather freely [49].

Extant studies report a positive association between longer diabetes duration and better performances of self-management among patients with type I [15] and type II [50] diabetes. A longer diabetes duration may result in an increased experience with self-care activities. Our results suggest that recently diagnosed patients with type II diabetes often lack the skills necessary for an efficient self-management of diabetes and therefore represent a special target group for DSME-programs.

Previous studies have emphasized that social support is important in the self-management of diabetes [16]. Family and friends constitute relevant support systems and are key motivators for patients with diabetes to stick to their self-management practices [23]. Especially individual commitments to family members are significant motivators to make better choices for a healthy lifestyle [51]. Our results are partly in line with the literature, as living together did not emerge as a significant facilitator of good SMB. Even though social relationships are relevant factors in sustaining motivation, living together with a spouse or a partner does not necessarily imply adherence in self-management. In this context it should be considered that elderly persons with diabetes represent a particularly vulnerable group who are especially dependent on family networks [52]. Second, instead of civic or partnership status, underlying characteristics such as quality of support, length and subjective satisfaction with the relationship may be more relevant concerning its effects on SMB [16].

Within the present analysis, age was positively associated with an increased probability for a mixed type of SMB, while both adherent and nonadherent patterns of SMB decreased with age. This accords to previous studies, pointing out that particularly older patients neglect various self-care practices, especially due to restricted physical abilities [23]. Furthermore, our analysis showed an increased risk for nonadherent SMB within younger groups. Concurrent with the results of McCarthy et al [13], targeting young persons who are at an early stage of the disease with regular self-management programs may be a reasonable strategy.

Various studies indicate that sex plays an important role in the adherence of self-management and female sex is reportedly associated with frequent SMB [17, 51]. The fact that our results, in contrast, do not reveal sex differences may be largely due to the fact that other studies operationalized the concept of SMB as self-care agency [17] or that these studies prospectively focused on the effects of sex-specific occupations [44]. Although our results do not indicate any relevant sex differences, DSME-programs should take into account the different needs of men and women in self-management of diabetes.

One last aspect regarding the composition of our latent classes concerns the indicator of "keeping a diabetes diary": 77.5% of the adherent, 13.9% of the non-adherent and 0.1% of the mixed adherent SMB pattern types keep a diabetes diary. Diabetes diaries are part of DSME for both patients with and without insulin treatment, both type I and type II [35]. In clinical routine, diabetes diaries are used to document SMBG or self-measurement of urine glucose and documentation of daily anti-diabetic medication. They are predominantly used for insulin dosage documentation and therefore more frequently used by patients with insulin treatment. A closer look at the mixed SMB pattern reveals that those consists of persons with diabetes who perform SMBG (47.0%), but are less likely to document these data in their diabetes diaries (0.1%). Our data therefore suggests that the comparatively low willingness to document one's own blood glucose within the mixed pattern might result from the belief that diabetes can be managed effectively without a diary. It is important to regard that we analysed the SMBG as a dichotomous (yes/no) variable for reasons of simplified data analysis within the framework of LCA. Keeping this in mind, another interpretation might be that both the mixed type (as well as the nonadherent type) choose to measure blood glucose on a more irregular basis, perhaps because they are on a therapy or metabolic situation which does not fulfil the criteria for reimbursement of self-assessment blood glucose test kits. Therefore, these respondents may choose not to document their SMBG but rather perform it on an occasional basis.

Composition of SMB Patterns and their corresponding factors remained largely unchanged within sensitivity analyses that excluded dietary plan and SMBG. Furthermore a majority of 66.5% of our respondents engaged in SMBG. This relatively high proportion of respondents performing self-assessment of blood glucose might be surprising, given that it is not a mandatory component of SMB for all persons with diabetes. As noted above (see "statistical analysis"), self-assessment of blood glucose is recommended and payed for only for certain patient groups with diabetes (namely those with insulin treatment and those with other antidiabetic medication with hypoglycaemia potential or certain medical situations at risk for an instable metabolism). However, patients are free to purchase test kits, for example in pharmacies, at their own expense and to use them to self-monitor their blood glucose. In concordance, literature indicates that self-assessment of blood glucose is used both by patients who do and who do not receive insulin as a treatment [53]. The proportion of self-assessments of blood glucose within our sample corresponds to a previous German national survey, showing that 62.8% of persons with diabetes use SMBG, which is in the same range as SMBG frequency in our study [24].

## Strengths and limitations

Contrary to other studies on self-management of diabetes, which rely predominantly on convenience samples or on RCTs within specific care settings, the GEDA-2014/2015-EHIS study is one of the few nationwide, population based surveys regarding this topic (see also GNHIES98 and DEGS1 [24, 54]; KORA-A [10, 55] is a further study from Germany but not nationwide). This is a strength of the present survey, as it provides data on diabetes SMB with generalizability to real world settings.

Another strength of our analyses is that the present survey covers all adult age groups. This is an added value in contrast to other studies conducted in the elderly (see Becker et al [10] or Murray et al [56]). Although the age distribution in the present survey is highly left-skewed, this does not imply that younger age cohorts are heavily underrepresented. Rather, these data correspond to the higher prevalence of type II diabetes and to the age-dependent diabetes prevalence provided by the German Diabetes Surveillance 2019 [38]. In addition, it should be noted that this age distribution was weighted and adjusted by the multi-stage survey design of

the GEDA Survey (and the corresponding weight-factors included in our regression models), thus minimizing a possible bias.

The present study adds an empirical typology of different types of SMB patterns based on population-based data to the existent literature. By employing latent class models, we identify coherent patterns of SMB by a data-appropriate method. In opposition to composite scores or the analysis of single item indicators, LCA takes account of the co-occurrences of various forms of SMB. Contrary to other operationalisations of diabetes self-management, we conceptualized adherence in SMB as a categorical, latent variable and used LCA to separate this construct from its corresponding measurement error.

Comparisons to results based on an additive index showed several advantages of this approach. For instance, one major advantage of latent class analysis is the imputation of missing values in the self-management indicators, while an additive index relies on listwise deletion of cases (in the present analyses, 77 cases had to be excluded that could otherwise be analysed within LCA). The low psychometric reliability of an adherence index indicates that SMB patterns cannot be ordered along a continuum (in the sense of a linear index), but instead form distinct categories. Estimating a latent multinomial variable in the context of an LCA forms an efficient strategy to statistically represent this basic assumption. An additive index also implies a parallel or tau-equivalent measurement model in which the individual indicators have an identical weight. This assumption seems questionable: is HbA1c measurement for instance replaceable by a participation in a retinopathy screening with respect to the operationalization of SMB adherence? Moreover, this assumption of an additive index complicates the interpretation of slope coefficients within corresponding regression models. Is a slope (i.e. the average difference between two groups regarding the additive index) of, say, one scale point of practical relevance? LCA, on the other hand, allows the results to be interpreted in terms of probability changes (for example: DSME participation increases the probability of an adherent SMB pattern by 51.7 percentage points). Additionally, this multivariate operationalization of SMB-patterns adapts better to everyday life compared to singular parameters because patients usually perform a variety of SMBs simultaneously. On the other hand, a major shortcoming of additive adherence-indexes lies in the fact that they neglect the co-occurrences of various forms of SMB.

Our study has various limitations, the most severe being the lack of information about respondents' HbA1c levels. Therefore, we cannot make any statements about the forms of SMB that meet the actual requirements of specific respondents (such as an actual need for optimization). Furthermore, the present data did not allow for a differentiation between respondents with type I or type II diabetes. Diabetes therapy and insulin treatment status, which are positively associated with various forms of self-monitoring and especially with SMBG [10], were not available with sufficient extent within the present survey. Against this background, it can be argued that not all self-management indicators used in the present analysis are appropriate for each respondent. Nonetheless, our results indicate that three coherent patterns of SMB emerge if respondents are pooled within one single sample. Furthermore, our typology of SMB-patterns refers only to actual and active practices of self-management among people with diabetes, regardless of diabetes therapy, diabetes type and HbA1c. Additionally, GEDA-EHIS 2014/2015 survey data does not distinguish between type I and II diabetes. However, it may be assumed that our sample predominantly consists of respondents with type II diabetes, given its high prevalence within the German population (7.0% to 7.4% [57]) and the comparatively low prevalence of type I diabetes (0.28% to 0.33% [58]). As our examined SMB variables are both applicable for type I and type II diabetes, we see no reasons that the potential diversity of the present sample may have led to difficulties with model identification. Nevertheless, an analysis stratified by these characteristics should be addressed by future research.

A further limitation concerns the assessment of SMB. The present survey employed self-reporting methods, which may be impacted by social desirability or a possible recall bias (likely resulting in an over-reporting of adherent SMB). Additionally, it may be argued that SMBG is not recommended and covered financially by health insurance companies for patients without insulin therapy. Since the status of insulin therapy was not available in the present survey for an analysis of subgroups, we addressed this issue in the context of a sensitivity analysis (see S3 Table, S3 Fig, S5 Table). Also, a classic diet plan in written form is not recommended in any German Diabetes Education programme, which gave reason to a second sensitivity analysis (see S4 Table, S4 Fig, S6 Table). However, the results of both additional analyses suggest that these limitations of our self-management assessments do not fundamentally affect or change the composition of the latent classes and their associated factors. Another limitation related to the assessment of SMB concerns the statistical consequences of combining binary and categorical dependent variables in the latent class analysis. Due to the different levels of measurement of the SMB indicators and to avoid sparse data, we necessarily had to rely on a dichotomization of the classification variables. There are two primary limitations associated with this: First, such a summary of variable characteristics is typically based on sample-specific parameters (e.g., median or mean). For this reason, classifications based on these summarized characteristics sometimes cannot be reproduced in other samples [59]. Second, MacCallum et al. [60] point out that the pooling of response categories can minimize differences among respondents such that the homogeneity of respondents within a formed group is overestimated.

The present analysis was limited to the associations of sociodemographic and disease-related factors with SMB. Research suggests that communication styles of health-care professionals as well as health provider–patient relations [51, 61] are relevant aspects in the implementation of self-management. However, the important role of the primary care physicians in supporting SMB is an aspect missing in our analysis, as well as non-family social support in general. However, up to our knowledge, we found no nationwide population-based study on SMB investigating communication styles of health-care professionals as confounder.

Another limitation concerns unavailable details on DSME participation, such as repetitive DSME participation, time gap between diabetes diagnosis and DSME participation, non-completion or cancellation of a DSME. Also, data on inscription of specialized chronic care management programs (DMPs) for participants with diabetes were not available. Among German patients who are inscribed in a DMP, there are local differences regarding DSME participation with proportions varying between 29.3% [62] and 41.1% [63]. Since enlistment in a DMP implies more frequent doctor/practice staff consultations than usual care, this variable constitutes a potential confounder that is associated with higher adherence in SMB.

Our results bear the limitations that are inherent of a cross-sectional study design. Associations between the different demographic and disease-related characteristics in the present study should be interpreted as mere descriptive correlations, rather than as causal relationships.

## Conclusions

Based on the co-occurrences of different SMB, we developed a parsimonious typology that distinguishes between three patterns. Subsequent latent variable regressions provide important insights for adequate tertiary prevention strategies within diabetes health care. Participation in a DSME-program showed to be the most relevant predictor of adherence in SMB, underlining the role of DSME in the treatment among patients with diabetes. Thus, there is arguably a need to ensure the chance to participate in a DSME program for each patient with diabetes. Our results imply that medical staff caring for persons with diabetes should be encouraged to

motivate their patients at the utmost to participate in a DSME. Our results also point toward groups with an increased risk of self-management deficits that should be given special consideration by health professionals, e.g. employed persons, persons with a short diabetes duration and persons with a low attendance towards their own health. Furthermore, fostering the mindfulness towards one's health might be a valuable strategy to increase efficacy in SMB. Further research and practical efforts should be undertaken to study how attentiveness toward patients' health could be increased.

## Supporting information

**S1 Fig. Profiles of the latent self-management classes in the 4-group model with 7 indicator variables.** Conditional prevalences, n = 1,466.
(TIF)

**S2 Fig. Profiles of the latent self-management classes in the 5-group model with 7 indicator variables.** Conditional prevalences, n = 1,466.
(TIF)

**S3 Fig. Results of the first sensitivity analysis (excluding self-measurement of blood glucose as indicator variable): Profiles of the latent self-management classes in the 3-group model with 6 indicator variables.** Conditional prevalences, n = 1,466.
(TIF)

**S4 Fig. Results of the second sensitivity analysis (excluding dietary plan as indicator variable): Profiles of the latent self-management classes in the 3-group model with 6 indicator variables.** Conditional prevalences, n = 1,466.
(TIF)

**S1 Table. Comparison of analysed net sample with complete cases (n = 1,466) versus respondents excluded from the analysis due to missing values in exogenous covariates (n = 241).**
(DOCX)

**S2 Table. Wordings for various questions regarding SMB within the GEDA questionnaire.**
(DOCX)

**S3 Table. Results of the first sensitivity analysis (excluding self-measurement of blood glucose as indicator variable): Information criteria (penalized likelihood criteria) for a series of weighted latent-class models without covariates (unconditional models).**
(DOCX)

**S4 Table. Results of the second sensitivity analysis (excluding dietary plan as indicator variable): Information criteria (penalized likelihood criteria) for a series of weighted latent-class models without covariates (unconditional models).**
(DOCX)

**S5 Table. Results of the first sensitivity analysis (excluding self-measurement of blood glucose as indicator variable): Multinomial latent variable regressions of posterior probabilities predicted by sociodemographic and disease-related factors (logistic slopes and marginal effects, latent classes parameters fixed within Manual ML-three-step approach).**
(DOCX)

**S6 Table. Results of the second sensitivity analysis (excluding dietary plan as indicator variable: Multinomial latent variable regressions of posterior probabilities predicted by**

**sociodemographic and disease-related factors (logistic slopes and marginal effects, latent classes parameters fixed within manual ML-three-step approach).**
(DOCX)

## Acknowledgments

We thank all participants of the GEDA-2014/2015-EHIS study. We would like to thank two anonymous reviewers for their valuable comments, which contributed significantly to the elaboration of the manuscript. We are grateful to Tatjana Steybe for proofreading the manuscript

## Author Contributions

**Conceptualization:** Marcus Heise, Astrid Fink, Jens Baumert, Christin Heidemann, Yong Du, Solveig Carmienke.

**Data curation:** Marcus Heise.

**Project administration:** Jens Baumert, Christin Heidemann, Yong Du.

**Writing – original draft:** Marcus Heise.

**Writing – review & editing:** Astrid Fink, Jens Baumert, Christin Heidemann, Yong Du, Thomas Frese, Solveig Carmienke.

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
