## [Decision Letter · Decision Letter 0]

5 Feb 2021

PONE-D-21-00720

Patterns and associated factors of diabetes self-management: Results of a latent class analysis in a German population-based study

PLOS ONE

Dear Dr. Heise,

Thank you for submitting your manuscript to PLOS ONE. After careful consideration, we feel that it has merit but does not fully meet PLOS ONE’s publication criteria as it currently stands. Therefore, we invite you to submit a revised version of the manuscript that addresses the points raised during the review process.

We look forward to receiving your revised manuscript.

Kind regards,

Antonio Palazón-Bru, PhD

Academic Editor

PLOS ONE

Reviewers' comments:

Reviewer's Responses to Questions

**Comments to the Author**

1. Is the manuscript technically sound, and do the data support the conclusions?

Reviewer #1: Yes

Reviewer #2: Partly

2. Has the statistical analysis been performed appropriately and rigorously? 

Reviewer #1: Yes

Reviewer #2: Yes

3. Have the authors made all data underlying the findings in their manuscript fully available?

Reviewer #1: Yes

Reviewer #2: Yes

4. Is the manuscript presented in an intelligible fashion and written in standard English?

Reviewer #1: Yes

Reviewer #2: No

5. Review Comments to the Author

Reviewer #1: This study provides a valuable contribution to the self-management literature. The premise of this study is novel and the use of LCA was appropriate and innovative with an adequate sample size. My primary concern with the use of words such as “deficient” and “compliant” throughout the manuscript, including in what you named your latent classes. These words are a condescending. Consider using words such as adherence and nonadherence.

Abstract: Was the regression a multinominal regression?

Word choice of compliant

Introduction:

Please justify why you included both TID and T2D in the sample as I would expect these two groups to have different patterns of self-management. Just the diversity of this sample may have lead to difficulty with model identification.

Line 71: do you mean diabetes self-efficacy? While generic self-efficacy is frequently cited in the literature it is not theoretically based.

Line 77: The work “appropriate” does not fit in this sentence.

Lines 98-101: list these are your inclusion/exclusion criteria to clarify that what you specially did for this study

Line 104-105: This sentence is not clear or a complete sentence “The outcome of our analysis are patterns of SMB among participants with diabetes in the last 12 months.”

It looks like you used a generic self-efficacy tool. This needs to be justified. I would expect to see a diabetes specific tool in this study and am not sure who a generic tool contributes to literature. I would suggest thinking about removing it.

Line 105: Please define what you mean by superordinate construct. This term is not regularly used in the LCA literature.

Line 123: What was your justification for using time since diagnosis of diabetes (≤ 2 years / >2 years) cutoff. This is also considering that most of your participants had diabetes longer than this and LCA performs better with indicators closer to 50%

Your LCA methods would be strengthened by adding information about how LCA is data-driven and how classes are mutually exclusive and exhaustive.

Line 326 : “proposal” Is the wrong word. This paper did not present a proposal but rather empiric grouping of people.

Line 338 : again please define “superordinate and abstract construct”

Lines 339- 341 “In a psychometric sense, one could argue that our categorical latent

340 variable represents a psychological trait (as a “personal commitment and receptiveness to

341 SMB”) rather than adherence in a clinical meaning.” This feels out of place in the discussion and was never addressed again. I am not sure your results supported this conclusion.

Line 430 “as it provides more authentic data on diabetes SMB.” Authentic is not the right word. Do you mean generalizable?

Reviewer #2: Diabetes self-management behaviour (SMB) usually concerns several different dimensions, which are often (additively) summarized as indexes. The present manuscript aims to identify certain patterns of SMB by conducting latent class analyses, and applies latent variable regressions to investigate the individual and disease-related characteristics of members of the different groups identified. The study is based on data from the German population-based GEA 2014/2105 study.

The topic covered in the paper is of high relevance for health policy makers who intend to increase compliance with self-management recommendations among people with diabetes. What I really like regarding the study is the statistical approach chosen. Especially, the latent class analysis proofs to be an innovative and helpful tool to overcome some of the shortcomings of SMB indexes previously utilized in the literature. Furthermore, the large population-based sample together with a reasonable set of individual information is an advantage of the present work compared to other studies.

However, from my point of view, the paper has limitations. The authors have already stated some of them in their “strengths and limitations” section. In the following, I will comment on major and minor points in the order of their appearance in the paper.

Major points

1. The paper considers a set of questions on different SMB. However, nothing is said about the exact wording, references are not given and comparisons to other instruments intended to elicit SMB are not mentioned.

2. Closely related to the first point, the introduction contains a short selection of different instruments to be found in the literature. What are the specific differences between these measures?

3. Point 1 also makes it impossible to get an idea about the improvement reached due to the latent class approach compared to other additive measures. One idea could be, to use the answers in the GEDA study to create an additive index and to compare corresponding findings on relevant factors considered in the second part of the analysis with results for this index. My guess would be that we could learn much more about the additive indexes and their limitations by such an approach.

4. Line 103: Assessment of SMB: The statistical consequences of combining binary and categorical dependent variables in the latent class analysis should be discussed.

5. As papers accepted by PLOS ONE are not copyedited by the journal, I would suggest to edit the paper. There are several linguistic inaccuracies in the text. E.g. line 63: add “regarding”, lines 118, 301: “sex” rather than “gender”, section “statistical analysis”. contains several repetitions of the expression “in order to”.

6. Line 93: A response rate of 26.9% is very low and gives reason for concern. Rightly, the author do not claim to have available representative but nationwide data. However, this response rate and the consequences for the interpretation of results should be more thoroughly discussed in the paper. If social desirability is an issue (see line 456), then this is also the case for the participation in the GEDA study itself.

7. Lines 167-173: It is argued that some behaviours are not part of “official” recommendations. Again, references would be helpful. Furthermore, it should be discussed, which behaviours are in fact recommended e.g. by any “German Diabetes Education program” (line 171)

8. Line 184: Interestingly, self-assessment of blood glucose is conducted by 66.4% of respondents although it is not recommend (see line 167). This is surprising and should be further discussed especially since this dimension is excluded later on.

9. Line 213: It is argued that further portioning does not provide added value in terms of interpretability. I am wondering how the results should have looked like to add value? The paper identifies three classes, which are obvious candidates from the very beginning. Hence, there seems to be tension to focus on three groups immediately. I think that the results with four or five classes should at least be briefly discussed in the paper even if they are discounted later on.

10. Lines 219-243: What is surprising and probably makes a difference is the proportion of respondents keeping a diabetes diary (77.7%, 13.9% and 0.1%, respectively). This should be discussed in more depth. What are the recommendations? Is this relevant only for some groups?

11. Lines 420-423: It is argued that gender (please use the term “sex” instead) plays an important role in other studies but not in the present one. Hence, the recommendation in the paper is that DSME programs should take “gender” differences into account. From this, the (somewhat malicious) general question emerges why the reader should consider the results of the paper anyway if other results in the literature seem to be more reliable. Maybe, a better way to deal with this observation is to discuss possible reasons for the absence of sex differences.

Minor points

12. In the abstract, more should be said about how SMB is measured in the paper.

13. Additionally, in the abstract it remains unclear what is meant by “compliant pattern” or “suboptimal self-management behaviour”.

14. Line 35: I suppose “employment” is part of the SES index. So why is employment incorporated as a single factor?

15. Line 78: “Few studies have investigated”. Please state which studies are relevant.

16. Lines 89ff.: Please also mention the decision of the Ethics Committee in the text (which is mentioned at the end of the paper)

17. Line 99: You report that 249 respondents have been removed due to missing sociodemographic or disease-related information. Have you considered imputing these values?

18. Line 117: Please discuss how the sociodemographic and disease-related variables have been selected. There are no references to any literature in this part of the paper.

19. Line 182: It should be stated what x-bar=3.9 means.

20. Line 270: Only 8.2% of those not having participated in a DSME program are classified as being “compliant”. I am wondering whether they can really be called “non-compliant” as they have probably not learned anything about recommendations. This point might also be discussed.

21. Line 294: The abbreviation “SM” has not been introduced before. Maybe this abbreviation should be avoided in any case?

22. Lines 336, 342: Results are five years old. Have DSME programs changed in the meantime?

23. Line 344: “Covering all adult age groups”. Is this really the case? Mean age is about 65 with a SD of 13.5. However, the lowest age is 19. Therefore, I guess that younger age cohorts are heavily underrepresented.

24. Line 441: Indeed, the study has limitations, but the authors should try avoid saying that they are “numerous”

6. PLOS authors have the option to publish the peer review history of their article (what does this mean?). If published, this will include your full peer review and any attached files.

Reviewer #1: No

Reviewer #2: No

---

## [Author Response · Author response to Decision Letter 0]

3 Mar 2021

Reviewer #1:

Comment: This study provides a valuable contribution to the self-management literature. The premise of this study is novel and the use of LCA was appropriate and innovative with an adequate sample size. 

Reply: Thank you very much for this comment. We are very pleased that you appreciate our statistical approach as an innovative way to identify patterns of diabetes self-management.

Comment: My primary concern with the use of words such as “deficient” and “compliant” throughout the manuscript, including in what you named your latent classes. These words are a condescending. Consider using words such as adherence and nonadherence.

Reply: We agree that the previous terminology could imply an (unintended) devaluation. For this reason, we have changed the labels of the latent classes as suggested. The revised manuscript refers to the latent classes as “adherent type” and “nonadherent type”. The words “deficient” and “compliant” have been replaced accordingly.

Comment: Abstract: Was the regression a multinominal regression?

Reply: Thank you for pointing this out. As described in our section on statistical analysis, we used multinomial latent variable regression to predict posterior probabilities of latent classes. In this context, Asparouhov and Muthén (2013) use the term of “multinomial latent variable regression” and distinguish it from a “regular” multinomial regression. We adopted this terminology and corrected the passage in the abstract (line: 28).

Comment: Abstract: Word choice of compliant

Reply: Please refer to our reply to your second comment above (regarding “deficient” and “compliant”.) 

Comment: Introduction: Please justify why you included both TID and T2D in the sample as I would expect these two groups to have different patterns of self-management. Just the diversity of this sample may have led to difficulty with model identification.

Reply: Thank you for this question. The questionnaire of the GEDA EHIS 2014/2015 survey does not distinguish between T1D and T2D, but merely asks participants whether they have ever been diagnosed with diabetes. Therefore, we could not distinguish between the two types of diabetes types in our data, as already pointed out in the limitations. Up to our knowledge, there are currently no published studies analysing patterns of self-management behaviour (SMB) stratified by type of diabetes (please also refer to the studies mentioned in our introduction, since none of those stratify by type the of diabetes). 

However, most SMB are necessary for both type 1 diabetes and type 2 diabetes, such as HbA1c-measurement, self-examination of feet, attendance of retinopathy-screenings and holding a diabetes pass. Furthermore, persons receiving insulin (both type 1 diabetes and type 2 diabetes) have similar recommendations regarding their SMB (such as the self-assessment of their blood glucose and keeping a diabetes-diary). We included this aspect in our revised introduction (lines: 57-59) and our discussion. (lines: 631-635).

According to the high prevalence of T2D the German population with 7.0 % to 7.4% (see Tönnies et al. 2019) in comparison to the low prevalence of Type 1 diabetes of 0.28% to 0.33% (see Goffrier 2017) it may be assumed that our sample predominantly consists of respondents with T2D. We also included this aspect in our revised limitations-section (lines: 627-631).

Comment: Line 71: do you mean diabetes self-efficacy? While generic self-efficacy is frequently cited in the literature it is not theoretically based.

Reply: We agree with your critique and completely removed the aspect of self-efficacy from the manuscript. Please also refer to your comment and our reply below (regarding “generic self-efficacy tool”). 

Comment: Line 77: The word “appropriate” does not fit in this sentence.

Reply: We agree that this phrasing was not correct. Instead, we wanted to point out that previous studies mostly neglected the co-occurrences of various forms of SMB and corrected the sentence accordingly.

Comment: Lines 98-101: list these are your inclusion/exclusion criteria to clarify that what you specially did for this study

Reply: As you suggested, we have marked this paragraph with the heading “inclusion/exclusion criteria”. Furthermore, we have restructured this paragraph and clarified our criteria for inclusion and exclusion. (lines: 122-132).

Comment: Line 104-105: This sentence is not clear or a complete sentence “The outcome of our analysis are patterns of SMB among participants with diabetes in the last 12 months.”

Reply: We shortened this sentence as follows: “The outcome of our analysis are patterns of SMB”. The aspect of “diabetes in the last 12 months” has been moved to the paragraph above on “inclusion criteria”. We hope that the sentence you objected to is now more comprehensible. (line: 135).

Comment: It looks like you used a generic self-efficacy tool. This needs to be justified. I would expect to see a diabetes specific tool in this study and am not sure who a generic tool contributes to literature. I would suggest thinking about removing it.

Reply: It is correct that we used a self-efficacy index without a direct relation to diabetes. Furthermore, this was the only self-efficacy inventory available in the GEDA-questionnaire. Our results with respect to this predictor do not allow us to relate to research literature. For this reason, we implemented your suggestion and completely removed the self-efficacy tool from our analyses. Please note that the exclusion of this predictor led to the inclusion of additional respondents, increasing our sample to n=1,466. All analyses were recalculated; tables and results have been corrected accordingly. In addition, the aspect of self-efficacy was completely removed from the introduction and the discussion.

Comment: Line 105: Please define what you mean by superordinate construct. This term is not regularly used in the LCA literature.

Reply: Originally, we were guided by the basic ideas of a factor analysis, according to which manifest items can be understood as indicators of a superordinate construct. However, we agree that this terminology is not common in the context of Latent Class Analysis. For this reason we have reworded the sentence.

Comment: Line 123: What was your justification for using time since diagnosis of diabetes (≤ 2 years / >2 years) cutoff. This is also considering that most of your participants had diabetes longer than this and LCA performs better with indicators closer to 50%.

Reply: Our original intent was that people with a recent diagnosis had not yet had the opportunity to participate in DSME training as DSME is usually undertaken shortly after diagnosis (although in routine care there might be some waiting time to get a placement). However, this concern turned out not to be true in subsequent analyses. Following your suggestion, we dichotomized the time since diagnosis at the respective median of 10 years. All results were corrected accordingly. Please note that time since diagnosis was only an exogenous covariate (not an indicator-variable) within the present analysis and therefore did not affect the composition of the latent classes.

Comment: Your LCA methods would be strengthened by adding information about how LCA is data-driven and how classes are mutually exclusive and exhaustive.

Reply: Thank you for this valuable addition, which we have included in the paragraph on statistical analysis. (lines: 189-192)

Comment: Line 326 : “proposal” Is the wrong word. This paper did not present a proposal but rather empiric grouping of people.

Reply: We have implemented your suggestion and rephrased the paragraph accordingly.

Comment Line 338: again please define “superordinate and abstract construct”

Reply: Please refer to your comment and our reply above (regarding “superordinate construct”). We have reworded the sentence.

Comment: Lines 339- 341 “In a psychometric sense, one could argue that our categorical latent 340 variable represents a psychological trait (as a “personal commitment and receptiveness to 341 SMB”) rather than adherence in a clinical meaning.” This feels out of place in the discussion and was never addressed again. I am not sure your results supported this conclusion.

Reply: We agree that this conclusion was of speculative nature and deleted this sentence.

Comment: Line 430 “as it provides more authentic data on diabetes SMB.” Authentic is not the right word. Do you mean generalizable?

Reply: Thank you for this comment. We replaced the word “authentic” with “generalizability to real world settings.”

 

Reviewer #2:

Comment: Diabetes self-management behaviour (SMB) usually concerns several different dimensions, which are often (additively) summarized as indexes. The present manuscript aims to identify certain patterns of SMB by conducting latent class analyses, and applies latent variable regressions to investigate the individual and disease-related characteristics of members of the different groups identified. The study is based on data from the German population-based GEA 2014/2105 study. The topic covered in the paper is of high relevance for health policy makers who intend to increase compliance with self-management recommendations among people with diabetes. What I really like regarding the study is the statistical approach chosen. Especially, the latent class analysis proofs to be an innovative and helpful tool to overcome some of the shortcomings of SMB indexes previously utilized in the literature. Furthermore, the large population-based sample together with a reasonable set of individual information is an advantage of the present work compared to other studies. However, from my point of view, the paper has limitations. The authors have already stated some of them in their “strengths and limitations” section. In the following, I will comment on major and minor points in the order of their appearance in the paper.

Reply: We thank you very much for this appreciation and for the numerous constructive comments that significantly help to improve our manuscript.

Major points

Comment: 1. The paper considers a set of questions on different SMB. However, nothing is said about the exact wording, references are not given and comparisons to other instruments intended to elicit SMB are not mentioned.

Reply: We provide the exact wording of these items as supporting information and refer to this supplemental table in the section “Assessment of SMB”. In the same paragraph, we compare our questions on different SMBs to the other instruments mentioned in the introduction. (lines: 134-153)

Comment: 2. Closely related to the first point, the introduction contains a short selection of different instruments to be found in the literature. What are the specific differences between these measures?

Reply: In the introduction, we have expanded the description of the four inventories that we mentioned. We point out differences in the items and wordings between these inventories. However, the most important argument to us is that all inventories assume a parallel or at least tau-equivalent measurement model. We have added this point in the revised manuscript. (lines: 62-81).

Comment: 3. Point 1 also makes it impossible to get an idea about the improvement reached due to the latent class approach compared to other additive measures. One idea could be, to use the answers in the GEDA study to create an additive index and to compare corresponding findings on relevant factors considered in the second part of the analysis with results for this index. My guess would be that we could learn much more about the additive indexes and their limitations by such an approach.

Reply: Thank you for this suggestion, which we have implemented accordingly. We describe the procedure in the section on “Statistical analysis” (lines: 216-220). At the end of the “Results” section, we have added a new subsection where we present the results of linear regression models (see in particular Table 4 and lines: 400-422). We discuss these results in the section “Strengths and limitations”. (lines: 595-614).

Comment: 4. Line 103: Assessment of SMB: The statistical consequences of combining binary and categorical dependent variables in the latent class analysis should be discussed.

Reply: While we fully agree that this is a valid argument, we have addressed this point in the “Limitations” section, where it better fits the line of reasoning. We added the following passage: “Due to the different levels of measurement of the SMB indicators and in order to avoid sparse data, we necessarily had to rely on a dichotomization of the classification variables. There are two primary limitations associated with this. First, such a summary of variable characteristics is typically based on sample-specific parameters (e.g., median or mean). For this reason, classifications based on these summarized characteristics sometimes cannot be reproduced in other samples (cf. Pastor et al. 2007). Second, MacCallum et al. (2002) point out that the pooling of response categories can minimize differences among respondents such that the homogeneity of respondents within a formed group is overestimated.” (lines: 647-654).

Comment: 5. As papers accepted by PLOS ONE are not copyedited by the journal, I would suggest to edit the paper. There are several linguistic inaccuracies in the text. E.g. line 63: add “regarding”, lines 118, 301: “sex” rather than “gender”, section “statistical analysis”. contains several repetitions of the expression “in order to”.

Reply: Thank you for these corrections, which we have implemented accordingly. We have made every effort to correct further linguistic inaccuracies in the text. Before resubmission, a linguistic expert revised the manuscript.

Comment: 6. Line 93: A response rate of 26.9% is very low and gives reason for concern. Rightly, the author do not claim to have available representative but nationwide data. However, this response rate and the consequences for the interpretation of results should be more thoroughly discussed in the paper. If social desirability is an issue (see line 456), then this is also the case for the participation in the GEDA study itself.

Reply: According to AAPOR (American Association for Public Opinion Research), 6 different response rates can be computed and reported while there is no single number or measure that reflects total survey quality. The response rate we reported here is the response rate 1 (RR1), which is also called the Minimum Response Rate according to AAPOR (https://www.aapor.org/AAPOR_Main/media/publications/Standard-Definitions20169theditionfinal.pdf). As a matter of fact, the refusal rate in GEDA 2014/15 is only 6.7% (see Lange et al. 2017). Furthermore, weights used in our analysis represent design weights (adjusting for sampling design) and adjustment weights (correcting deviations between study population and German national population statistics) (see Lange et al. 2017). This weighting approach enables the distribution of age, sex, education and region of the sample to be equal to that of the German national adult population and thus our results can be considered nationally representative with respect to these characteristics. In spite of the relatively low Minimum Response Rate, a selection bias with respect to other characteristics cannot be excluded, however weighting by age, sex, education and region addresses the main sources of deviations between population-based surveys and the general population. 

Social desirability regarding study participation cannot be fully excluded, however it may be more an issue in samples where study participant and study team may know each other (e.g. in- or outpatient settings) which may contribute to the willingness of participation. The present study, in contrast, was drawn randomly from the general population with no personal relation between participant and study team before beginning of the study, thus an association of social desirability and participation is rather unlikely here.

Comment: 7. Lines 167-173: It is argued that some behaviours are not part of “official” recommendations. Again, references would be helpful. Furthermore, it should be discussed, which behaviours are in fact recommended e.g. by any “German Diabetes Education program” (line 171)

Reply: Thank you for this comment. We may have used a misleading choice of words. We clarified our formulation in the section “Statistical analysis” (in the paragraph introducing the sensitivity analyses; (lines: 224-238) and included the requested citations. All SMB we examined are covered and recommended by the current diabetes therapy and DSME guideline with two minor exceptions: Self-assessment of blood glucose is only financially covered and recommended for those patients receiving insulin or oral antidiabetic medication with hypoglycaemia risk or who are in instable metabolic situations (e.g. undergoing an operation). Secondly, DSME program guidelines recommend developing a dietary concept but not strictly a diet plan. The usage of strict diet plans has decreased in routine clinic care for therapy of diabetes over the last years, therefore, we choose to run a sensitivity analyses for both variables “self-assessment of blood glucose" and “diet plan". 

Comment: 8. Line 184: Interestingly, self-assessment of blood glucose is conducted by 66.4% of respondents although it is not recommend (see line 167). This is surprising and should be further discussed especially since this dimension is excluded later on.

Reply: Thank you for your comment. We apologize for our choice of words, which may have been misleading. As we elaborated in the answer to Comment 7 and in the revised section on “Statistical analysis”, self-assessment of blood glucose is indeed recommended and payed for, however only for certain patient groups with diabetes (namely those with insulin treatment and those with other antidiabetic medication with hypoglycaemia potential or certain medical situations at risk for an instable metabolism). However, patients are free to purchase test kits in the pharmacy at their own expense and use this to self-monitor their blood glucose. In concordance, literature indicates that self-assessment of blood glucose is used both by patients who do and who do not receive insulin as a treatment (see Klonoff et al. 2011). The percentage of self-assessments of blood glucose in our sample corresponds to a previous German national survey, showing that 62.8% of persons with diabetes use SMBG, which is in the same range as SMBG frequency in our study (see Du et al. 2015). We included this aspect in the last paragraph of our “Main findings compared to other population-based studies”-section (lines: 556-570).

Please remark that we did not exclude self-assessment of blood glucose from the analyses. However, we did perform a sensitivity analyses for SMB patterns via LCA as we are aware that self-assessment of blood glucose is a SMB criteria which is not recommended for every patient with diabetes. However, our LCA pattern remained stable even when not taking self-assessment of blood glucose as an SMB-indicator into account.

Comment: 9. Line 213: It is argued that further portioning does not provide added value in terms of interpretability. I am wondering how the results should have looked like to add value? The paper identifies three classes, which are obvious candidates from the very beginning. Hence, there seems to be tension to focus on three groups immediately. I think that the results with four or five classes should at least be briefly discussed in the paper even if they are discounted later on.

Reply: We completely agree with you that the selection of the 3-class solution requires justification. However, in the original manuscript we were concerned that the description of the discarded partitionings might be perceived as redundant. In the revised manuscript, we have described the 4-class and 5-class partitionings (and the reasons for rejecting them) in more detail. (lines: 282-293)

Comment: 10. Lines 219-243: What is surprising and probably makes a difference is the proportion of respondents keeping a diabetes diary (77.7%, 13.9% and 0.1%, respectively). This should be discussed in more depth. What are the recommendations? Is this relevant only for some groups?

Reply: Thank you for this comment. Indeed, 77.5% participants (results of new analyses) of the adherent, 13.9 % of the non-adherent and 0.1 % of the mixed adherent SMB pattern types keep a diabetes diary. Diabetes diaries are part of DSME for both patients with and without insulin treatment, both T1DM and T2DM (see AkdÄ 2011). In clinical routine, diabetes diaries are used to document self-assessment of blood glucose or self-measurement of urine glucose and documentation of daily anti-diabetic medication, predominantly for insulin dosage and therefore are more used by patients with these characteristics. It is worth mentioning that urine glucose measurement is used for monitoring glycaemic control for patients without insulin treatment, but it is not a widely applied in Germany. It was not an investigated variable in the GEDA 2014/2015 EHIS survey. A closer look at the mixed SMB pattern reveals, that those are persons with diabetes who perform self-assessment of blood glucose (47.0%), but are less likely to document these data in their diabetes diaries (1.0%). Our data therefore suggests that the comparatively low willingness to document one’s own blood glucose might result from the belief that diabetes can be managed effectively without a diary. Please note we analysed the self-assessment of blood glucose measurement as a dichotomous (yes/no) variable within LCA framework, therefore neglecting a metric for precise frequency. Keeping this in mind, another interpretation might be that those are patients who choose to measure blood glucose on a more irregular basis, perhaps because they are on a therapy or metabolic situation which does not fulfil the criteria for reimbursement of self-assessment test kits. Therefore, they may choose not to document their self-assessment of blood glucose, but to perform it from time to time to know that their diabetes is well managed and therefore refrain from using the diabetes diary. We included this aspect in the paragraph of our Discussion on “Main findings” (lines: 538-555).

Comment 11. Lines 420-423: It is argued that gender (please use the term “sex” instead) plays an important role in other studies but not in the present one. Hence, the recommendation in the paper is that DSME programs should take “gender” differences into account. From this, the (somewhat malicious) general question emerges why the reader should consider the results of the paper anyway if other results in the literature seem to be more reliable. Maybe, a better way to deal with this observation is to discuss possible reasons for the absence of sex differences.

Reply: As suggested, the revised manuscript uses the term “sex”. We attribute these deviations primarily to different research questions and to divergent operationalizations of SMB. We have addressed this aspect accordingly in the revised manuscript (lines: 531-537). Nevertheless, we believe that our findings should not be used as a rationale to negate sex-specific strategies in DSME training. For this reason, we argue that the revised manuscript should also refer to the specific role of sex, which has been highlighted in other studies.

 

Minor points

Comment 12. In the abstract, more should be said about how SMB is measured in the paper.

Reply: We revised the abstract and added information regarding the measurement of SMB (lines: 26-28).

Comment 13. Additionally, in the abstract it remains unclear what is meant by “compliant pattern” or “suboptimal self-management behaviour”.

Reply: We revised the abstract and added descriptions of the latent classes (lines: 32-37).

Comment 14. Line 35: I suppose “employment” is part of the SES index. So why is employment incorporated as a single factor?

Reply: Please note that for unemployed or retired respondents, the SES index does not refer to the current employment status, but uses the last professional activity for its calculation instead. However, the current occupational status constitutes another important predictor with regard to SMB (instead of the last professional activity). Furthermore, the SES index also refers to education and income. We found no evidence of multicollinearity between the SES index and occupational status and thus use both as two separate factors. We addressed the point you raised in the section on "Sociodemographic and disease-related variables." (lines: 169-174)

Comment 15. Line 78: “Few studies have investigated”. Please state which studies are relevant.

Reply: Apart from the study from Mc Carthy et al., which was already mentioned in the manuscript, we could identify only two additional studies of relevance that we included in the revised manuscript (lines: 83-86). All other studies known to us either use additive scores or single items for measurement of SMB. 

Comment 16. Lines 89ff.: Please also mention the decision of the Ethics Committee in the text (which is mentioned at the end of the paper)

Reply: We included the decision of the German Federal Commissioner for Data Protection and Freedom of Information in the corresponding paragraph (lines: 117-121). We also added the following information: “All respondents gave their written informed consent. Participants were informed about the goals and contents of the study, about privacy and data protection proceedings, and that their participation in the study was voluntary.”

Comment 17. Line 99: You report that 249 respondents have been removed due to missing sociodemographic or disease-related information. Have you considered imputing these values?

Reply: We compare the SMB-outcomes between the removed respondents and the excluded respondents. These results are provided as a supplemental table in the section “Inclusion / exclusion criteria” (lines: 128-132 and S1 Table) With respect to the indicator variables of latent SMB patterns, we find no difference between the two groups. Little and Rubin (2014; cited in the revised manuscript) point out that in such a constellation the risks of imputation may outweigh its benefits. With this in mind, we decided against imputing the exogenous variables. Moreover, missing values of the SMB indicators are imputed anyway within the ML estimation algorithm of the LCA. We did not want to convolute this ML-based imputation procedure by additionally imputing for exogenous covariates. Please also note that the number of removed respondents decreased to n=241 because the exogenous variable of self-efficacy has been excluded from the analysis (see Reviewer#1)

Comment 18. Line 117: Please discuss how the sociodemographic and disease-related variables have been selected. There are no references to any literature in this part of the paper.

Reply: We addressed this issue as follows and included all the references in the corresponding passage: “The selection of covariates was based on the availability of data in the GEDA survey and on central findings of previous studies. Among sociodemographic factors of SMB, age [Ruggiero et al. 1997; Matricciani et al. 2015], sex [Carter et al. 1998], socioeconomic status [Adu et al. 2019; Campbell et al. 2017], employment [Adu et al. 2019], and partnership status [Haines et al. 2018] have been reported in the literature. Among disease-related variables, time since diagnosis [Adu et al. 2019], DSME-participation [Becker et al. 20202; Carmienke et al. 2020], limitation due illness [Du et al. 2015] and attendance toward health [Adu et al. 2019; Lange et al. 2017] emerged as factors of SMB. We adopted these variables as exogenous covariates within our analyses. However, the selection of these covariates was not based on a systematic literature review. Thus, our analysis does not aim at a systematic comparison of theories, but rather has an explorative character, which is primarily due to the availability of the data within the present survey.” (lines: 156-165)

Comment 19. Line 182: It should be stated what x-bar=3.9 means.

Reply: Originally, we used this to refer to the mean value of the self-efficacy scale. However, since this instrument was excluded from the analyses (as recommended by Reviewer 1), the sentence you mentioned was also deleted from the manuscript.

Comment 20. Line 270: Only 8.2% of those not having participated in a DSME program are classified as being “compliant”. I am wondering whether they can really be called “non-compliant” as they have probably not learned anything about recommendations. This point might also be discussed.

Reply: We apologise if we misunderstand your comment here, as you refer to both compliant and non-compliant respondents. We interpret your comment to mean that people can be compliant or adherent even if they have not taken part in a DSME training course. We added this point into the discussion where we thought it would fit better into the argument. In the first paragraph of the section “Main findings compared to other population-based studies” we have added the following passage: “We find a strong association between DSME-participation and adherent SMB. However, based on average posteriori probabilities, 12.1% of those not having participated in a DSME program are classified as being “adherent”.Thus, non-participants can also be adherent in their SMB and instead use alternative ways to inform themselves about coping with diabetes or receive extensive support from the practice personnel. In addition, many reasons exist for (adherent) respondents to decide against DSME training. In particular, truck drivers or commuters may not be able to attend a DSME program due to work-related reasons. Regardless, our results indicate that people with diabetes should be informed by their general practitioner about the possibility of DSME training.” (lines: 462-472). If we have misunderstood your remark, please give us the opportunity to correct our mistake.

Comment 21. Line 294: The abbreviation “SM” has not been introduced before. Maybe this abbreviation should be avoided in any case?

Reply: This was indeed a typo. The abbreviation should actually refer to self-management behavior (SMB) and has been corrected accordingly.

Comment 22. Lines 336, 342: Results are five years old. Have DSME programs changed in the meantime?

Reply: Even though our data are five years old, we are of the opinion that they are still of high relevance, especially since SMB is rarely assessed in nation-wide surveys. DSME programs, their curricula, aims and content have not changed in the last five years. However, new medications emerging, as SGLT2 inhibitors, have been integrated in the medication education sessions. For specific new technical devices, e.g. as application and use of continuous subcutaneous blood glucose measurement a new DSME program for teaching use and boundaries of this technique has been developed. However, this latter program is used as an addition to the other DSME. Also, in the current pandemic, Online-DSME formats have emerged to ensure some stability of care. However, they are not widely applied, as 74.7 % of registered DSME trainers reported not ever having performed an Online-DSME and educate the same curriculum as the “vis-à-vis” DSME. (Deutsche Diabetes-Hilfe 2021). We included this aspects in the revised section of the “Discussion” (lines: 442-451).

Comment 23. Line 344: “Covering all adult age groups”. Is this really the case? Mean age is about 65 with a SD of 13.5. However, the lowest age is 19. Therefore, I guess that younger age cohorts are heavily underrepresented.

Reply: We deleted the phrase “Covering all adult age groups” from the first section of the discussion and moved it to the section “strength and limitations”, where we addresses this issue in more detail (lines: 579-587). We have added the following paragraph: “Another strength of the present survey is that the GEDA study covers all adult age groups. This is an added value to comparable studies (see Becker et al [2020] or Murray et al [2016]). Although the age distribution in the present survey is highly left-skewed, this does not imply that younger age cohorts are heavily underrepresented. Rather, these data correspond to the higher prevalence of type 2 diabetes and to the age-dependent diabetes prevalence provided by the German Diabetes Surveillance 2019 (cf. Scheidt-Nave et al. 2019). In addition, it should be noted that this age distribution was weighted and adjusted by the multi-stage survey design of the GEDA Survey (and the corresponding weight-factors included in our regression models), thus minimizing a possible bias. (Deutsche Diabetes-Hilfe 2021)

Comment: 24. Line 441: Indeed, the study has limitations, but the authors should try avoid saying that they are “numerous”

Reply: We replaced “numerous limitations” with “various limitations”

References:

Adu MD, Malabu UH, Malau-Aduli AEO, Malau-Aduli BS. Enablers and barriers to effective diabetes self-management: A multi-national investigation. PLoS One. 2019; 14:e0217771. doi: 10.1371/journal.pone.0217771 PMID: 31166971.

Arzneimittelkommission Der Deutschen Ärzteschaft (AkdÄ), ABDA-Bundesvereinigung Deutscher Apothekerverbände, Deutsche Diabetes-Gesellschaft (DDG), Deutsche Gesellschaft Für Allgemeinmedizin Und Familienmedizin (DEGAM), Fachkommission Diabetes Der Sächsischen Landesärztekammer (FKDS), Verband Der Diabetesberatungs- Und Schulungsberufe Deutschland (VDBD), et al. Nationale VersorgungsLeitlinie Diabetes - Strukturierte Schulungsprogramme - Kurzfassung, 1. Auflage. Bundesärztekammer (BÄK); Kassenärztliche Bundesvereinigung (KBV); Arbeitsgemeinschaft der Wissenschaftlichen Medizinischen Fachgesellschaften (AWMF); 2012.

Arzneimittelkommission Der Deutschen Ärzteschaft (AkdÄ), Deutsche Diabetes Gesellschaft (DDG), Deutsche Gesellschaft Für Allgemeinmedizin Und Familienmedizin (DEGAM), Deutsche Gesellschaft Für Innere Medizin (DGIM) (Vertreten Durch Die DDG), Verband Der Diabetesberatungs- Und Schulungsberufe Deutschland (VDBD), Ärztliches Zentrum Für Qualität In Der Medizin (ÄZQ). Nationale VersorgungsLeitlinie Therapie des Typ-2-Diabetes - Langfassung, 1. Auflage. Bundesärztekammer (BÄK); Kassenärztliche Bundesvereinigung (KBV); Arbeitsgemeinschaft der Wissenschaftlichen Medizinischen Fachgesellschaften (AWMF); 2013.

Asparouhov T, Muthén B. Auxiliary Variables in Mixture Modeling: Three-Step Approaches Using M plus. Structural equation modeling: A multidisciplinary Journal. 2014; 21:329–41. doi: 10.1080/10705511.2014.915181.

Becker J, Emmert-Fees KMF, Greiner GG, Rathmann W, Thorand B, Peters A, et al. Associations between self-management behavior and sociodemographic and disease-related characteristics in elderly people with type 2 diabetes - New results from the population-based KORA studies in Germany. Prim Care Diabetes. 2020. doi: 10.1016/j.pcd.2020.01.004 PMID: 32088161.

Campbell DJT, Manns BJ, Hemmelgarn BR, Sanmartin C, Edwards A, King-Shier K. Understanding Financial Barriers to Care in Patients With Diabetes. The Diabetes Educator. 2017; 43:78–86. doi: 10.1177/0145721716679276. PMID: 27920081.

Carmienke S, Baumert J, Gabrys L, Heise M, Frese T, Heidemann C, et al. Participation in structured diabetes mellitus self-management education program and association with lifestyle behavior: results from a population-based study. BMJ Open Diabetes Res Care. 2020; 8. doi: 10.1136/bmjdrc-2019-001066 PMID: 32205327.

Carter PA. Self-care agency: the concept and how it is measured. J Nurs Meas. 1998; 6:195–207.

Deutsche Diabetes-Hilfe. Deutscher Gesundheitsbericht 2021. Die Bestandsaufnahme. Mainz: Kirchheim; 2020.

Du Y, Heidemann C, Schaffrath Rosario A, Buttery A, Paprott R, Neuhauser H, et al. Changes in diabetes care indicators: findings from German National Health Interview and Examination Surveys 1997-1999 and 2008-2011. BMJ Open Diabetes Res Care. 2015; 3:e000135. doi: 10.1136/bmjdrc-2015-000135 PMID: 26629347.

Goffrier B, Schulz Mandy, Bätzing-Feigenbaum J. Administrative Prävalenzen und Inzidenzen des Diabetes mellitus von 2009 bis 2015. Zentralinstitut für die kassenärztliche Versorgung in Deutschland (Zi); 2017.

Haines L, Coppa N, Harris Y, Wisnivesky JP, Lin JJ. The Impact of Partnership Status on Diabetes Control and Self-Management Behaviors. Health Educ Behav. 2018; 45:668–71. doi: 10.1177/1090198117752783 PMID: 29361845.

Klonoff DC, Blonde L, Cembrowski G, Chacra AR, Charpentier G, Colagiuri S, et al. Consensus report: the current role of self-monitoring of blood glucose in non-insulin-treated type 2 diabetes. J Diabetes Sci Technol. 2011; 5:1529–48. Epub 2011/11/01. doi: 10.1177/193229681100500630 PMID: 22226276.

Lange C, Finger JD, Allen J, Born S, Hoebel J, Kuhnert R, et al. Implementation of the European health interview survey (EHIS) into the German health update (GEDA). Archives of Public Health. 2017; 75:40.

Little RJA, Rubin DB. Statistical Analysis with Missing Data. 2nd ed. New York, NY: John Wiley & Sons; 2014.

MacCallum RC, Zhang S, Preacher KJ, Rucker DD. On the practice of dichotomization of quantitative variables. Psychol Methods. 2002; 7:19–40. doi: 10.1037/1082-989x.7.1.19 PMID: 11928888.

Matricciani L, Jones S. Who cares about foot care? Barriers and enablers of foot self-care practices among non-institutionalized older adults diagnosed with diabetes: an integrative review. The Diabetes Educator. 2015; 41:106–17.

Murray CM, Shah BR. Diabetes self-management education improves medication utilization and retinopathy screening in the elderly. Prim Care Diabetes. 2016; 10:179–85. Epub 2015/11/25. doi: 10.1016/j.pcd.2015.10.007 PMID: 26620389.

Pastor DA, Barron KE, Miller BJ, Davis SL. A latent profile analysis of college students’ achievement goal orientation. Contemporary Educational Psychology. 2007; 32:8–47. doi: 10.1016/j.cedpsych.2006.10.003.

Ruggiero L, Glasgow R, Dryfoos JM, Rossi JS, Prochaska JO, Orleans CT, et al. Diabetes self-management. Self-reported recommendations and patterns in a large population. Diabetes Care. 1997; 20:568–76. doi: 10.2337/diacare.20.4.568 PMID: 9096982.

Scheidt-Nave C, editor. Diabetes in Deutschland. Bericht der Nationalen Diabetes-Surveillance 2019. Berlin: Robert Koch-Institut; 2019.

Tönnies T, Röckl S, Hoyer A, Heidemann C, Baumert J, Du Y, et al. Projected number of people with diagnosed Type 2 diabetes in Germany in 2040. Diabet Med. 2019; 36:1217–25. Epub 2019/02/13. doi: 10.1111/dme.13902 PMID: 30659656.

---

## [Editor Report · Decision Letter 1]

10 Mar 2021

Patterns and associated factors of diabetes self-management: Results of a latent class analysis in a German population-based study

PONE-D-21-00720R1

Dear Dr. Heise,

We’re pleased to inform you that your manuscript has been judged scientifically suitable for publication and will be formally accepted for publication once it meets all outstanding technical requirements.

Kind regards,

Antonio Palazón-Bru, PhD

Academic Editor

PLOS ONE

Additional Editor Comments (optional): I have analyzed all the reviewers' concerns and their responses and I agree with all of them. Therefore, I recommend the acceptance of the paper in its current form in PLoS One. Congratulations!
---

## [Editor Report · Acceptance letter]

12 Mar 2021

PONE-D-21-00720R1 

Patterns and associated factors of diabetes self-management: Results of a latent class analysis in a German population-based study 

Dear Dr. Heise:

I'm pleased to inform you that your manuscript has been deemed suitable for publication in PLOS ONE. Congratulations! Your manuscript is now with our production department. 

Kind regards, 

on behalf of

Dr. Antonio Palazón-Bru 

Academic Editor

PLOS ONE